# Functional Dizziness as a Spatial Cognitive Dysfunction

**DOI:** 10.3390/brainsci14010016

**Published:** 2023-12-23

**Authors:** Hayo A. Breinbauer, Camilo Arévalo-Romero, Karen Villarroel, Claudio Lavin, Felipe Faúndez, Rosario Garrido, Kevin Alarcón, Ximena Stecher, Francisco Zamorano, Pablo Billeke, Paul H. Delano

**Affiliations:** 1Laboratory for Clinical Neuro-Otology and Balance-Neuroscience, Department of Neuroscience, Facultad de Medicina, Universidad de Chile, Santiago 8331150, Chile; camiloarevaloromero@gmail.com (C.A.-R.); kvillarroel@ug.uchile.cl (K.V.); f.faundezj@gmail.com (F.F.); rogarchar@gmail.com (R.G.); kevin.alarcon@ug.uchile.cl (K.A.); pdelano@uchile.cl (P.H.D.); 2Department of Otolaryngology, Facultad de Medicina Clínica Alemana, Universidad del Desarrollo, Santiago 7610615, Chile; 3Laboratorio de Neurociencia Social y Neuromodulación, Centro de Investigación en Complejidad Social (neuroCICS), Facultad de Gobierno, Universidad del Desarrollo, Santiago 7610615, Chilepbilleke@udd.cl (P.B.); 4Department of Radiology, Facultad de Medicina Clínica Alemana, Universidad del Desarrollo, Santiago 7610615, Chile; xstecher@alemana.cl (X.S.); francisco.zamorano@uss.cl (F.Z.); 5Facultad de Ciencias para el Cuidado de la Salud, Universidad San Sebastián, Santiago 8420524, Chile; 6Centro Avanzado de Ingeniería Eléctrica y Electrónica, AC3E, Universidad Técnica Federico Santa María, Valparaíso 2390123, Chile; 7Servicio de Otorrinolaringología, Hospital Clínico Universidad de Chile, Santiago 8380456, Chile

**Keywords:** persistent postural perceptual dizziness, functional dizziness, spatial navigation, spatial cognition, functional neurological disorder, chronic dizziness, chronic vertigo

## Abstract

(1) Background: Persistent postural-perceptual dizziness (PPPD) is a common chronic dizziness disorder with an unclear pathophysiology. It is hypothesized that PPPD may involve disrupted spatial cognition processes as a core feature. (2) Methods: A cohort of 19 PPPD patients underwent psycho-cognitive testing, including assessments for anxiety, depression, memory, attention, planning, and executive functions, with an emphasis on spatial navigation via a virtual Morris water maze. These patients were compared with 12 healthy controls and 20 individuals with other vestibular disorders but without PPPD. Vestibular function was evaluated using video head impulse testing and vestibular evoked myogenic potentials, while brain magnetic resonance imaging was used to exclude confounding pathology. (3) Results: PPPD patients demonstrated unique impairments in allocentric spatial navigation (as evidenced by the virtual Morris water maze) and in other high-demand visuospatial cognitive tasks that involve executive functions and planning, such as the Towers of London and Trail Making B tests. A factor analysis highlighted spatial navigation and advanced visuospatial functions as being central to PPPD, with a strong correlation to symptom severity. (4) Conclusions: PPPD may broadly impair higher cognitive functions, especially in spatial cognition. We discuss a disruption in the creation of enriched cognitive spatial maps as a possible pathophysiology for PPPD.

## 1. Introduction

Persistent postural-perceptual dizziness (PPPD) is the leading cause of chronic vestibular syndrome, characterized by constant non-vertiginous dizziness, unsteadiness, and sensations of swaying or rocking [1,2,3,4]. These symptoms, for the majority of the day, on most days, significantly impair the quality of life in patients. For an in-depth understanding of PPPD’s epidemiology, diagnostics, and clinical aspects, see Staab 2023 [5]. As outlined there, the pathophysiology of PPPD is still under active investigation, to which this study contributes.

Currently, PPPD is categorized as a functional disorder, as no definitive structural lesions are evident in either the vestibular organs or the central nervous system [6]. The most current pathophysiological model suggests that a triggering event—often another vestibular disorder but sometimes a non-vestibular issue like an acute anxiety episode—leads to a cascade of poorly understood functional neural adaptations. These encompass alterations in the processing of sensory information and vestibular responses within the brain, rather than overt structural changes [5,6,7,8].

A current model hypothesizes that PPPD may involve functional disruptions at the cortical network level, affecting systems related to motion perception, postural control, locomotion, and spatial orientation [9,10,11,12,13,14,15,16,17,18,19,20,21,22,23,24]. Such disruptions could potentially contribute to the characteristic dizziness observed in PPPD [5], and may also manifest as heightened attention to motion [9], misperception of movement [25], altered posture [24], increased effort in postural control [26], heightened dependence on visual input [27], and compromised spatial navigation [12], among others [5]. It is important to note that while these associations are suggested, the exact pathophysiological mechanisms underlying these symptoms in PPPD remain to be conclusively determined.

Various theories attempt to elucidate the functional neural adaptations in PPPD. One perspective proposes that PPPD interferes with the mechanisms crucial for maintaining an accurate internal representation of external space [6,10,12,28]. Rooted in new models, this perspective combines sensory perception, threat evaluation, and cognitive assessment of surrounding space into an integrated multimodal system [28,29,30,31,32,33,34,35]. This system actively creates context-sensitive (or risk-adaptive) vestibulo-spatial and vestibulo-temporal maps of one’s environment [28].

The brain’s ability to quickly create these maps is crucial for maintaining balance and swiftly adapting to the ever-changing environment. This process helps prevent falls and is dependent on the brain’s fundamentally multimodal and integrative method of processing vestibular information [36].

According to some authors, no cortical regions are exclusively unimodal [37]. Yet, some senses such as vision and hearing process their initial inputs in dedicated primary cortices, which then relay the information to other areas of the brain. The concept of a ‘vestibular cortex’ in humans has been explored, with areas such as the parieto-insular vestibular cortex (PIVC) proposed as primary regions for vestibular input processing [38]. However, there is growing evidence that vestibular processing is characterized by a diffuse network. This network is a complex, interwoven set of multimodal networks that directly engage with the vestibular nuclei and project to multiple cortical regions [39,40,41,42].

Furthermore, there is mounting evidence to suggest that these extensive vestibular networks are not just confined to what might traditionally be labeled as the ‘vestibular domain’, such as body positioning, movement, and broader spatial navigation [43,44]. They also encompass a range of other cognitive and autonomic functions including, but not limited to, pain processing, emotional regulation, and executive functions [41,45,46].

In adherence to the principle ‘Where there is function, there is potential for dysfunction’, we posit that dysfunctions in the brain’s ability to merge vestibular inputs with other sensory information and cognitive assessments may underlie PPPD. Such a disruption in the integration process could impair the construction of accurate, context-sensitive perceptual maps, which are crucial for interpreting the external world. Furthermore, we suggest that this disruption may manifest as a persistent mismatch between an error-prone, cognitively enriched perceptual map and the actual sensory afferences. This misalignment may contribute, at least in part, to the persistent dizziness that is characteristic of PPPD.

Therefore, our study investigates spatial navigation and cognitive functions in PPPD patients by actively comparing them with healthy individuals and those with other vestibular disorders but without PPPD. This approach aims to delineate PPPD-specific cognitive impairments, particularly in tasks requiring advanced spatial processing skills.

## 2. Materials and Methods

A cross-sectional study was conducted involving three age-matched groups of subjects: (i) patients diagnosed with PPPD; (ii) patients diagnosed with vestibular disorders other than PPPD; and (iii) healthy volunteers.

The non-PPPD vestibular disorders encompassed benign paroxysmal positional vertigo (BPPV), vestibular neuritis, vestibular migraine (VM), and Ménière’s disease. These disorders were selected because they represent the most common non-PPPD conditions in neuro-otology and result in various types of vestibular dysfunctions. BPPV patients were evaluated before undergoing repositioning maneuvers, while Ménière’s and migraine patients were assessed during inter-ictal periods. Patients diagnosed with vestibular neuritis were only included if they were assessed at least 3 months post-onset, displayed no spontaneous nystagmus, and had not started vestibular rehabilitation by the time of the study procedures.

We recognize the diagnostic intricacies arising from the symptomatic intersection of VM and PPPD. To accurately represent the clinical spectrum, we included VM in both PPPD and non-PPPD cohorts, mirroring its prevalence and the clinical realities encountered in practice. To distinguish between the two, we employed rigorous criteria primarily based on the temporal pattern of vestibular symptoms. For VM, we mandated the presence of discrete episodes with definitive onsets and cessations, and minimal interictal manifestations, requiring at least half of the episodes to include headache or other cardinal VM symptoms [47]. For PPPD, we stipulated continuous symptoms, pervasive throughout most of the day and on most days, clearly segregating any overlaid VM episodes [4].

The study solicited participation from patients attending the outpatient neurotology–otolaryngology unit at Clínica Alemana de Santiago medical center in Chile from January 2022 to September 2023. Conducted in alignment with the Helsinki declaration, the research received approval from our center’s Ethical Committee (Approval number UIEC 1081). All participants provided written informed consent. Eligibility criteria required participants to be between 18 and 65 years.

Initial medical consultations for both PPPD and other vestibular diseases adhered to the 2023 Bárány Society diagnostic criteria for definitive disease diagnoses [4,47,48,49,50]. After diagnosis determination, examiners conducting assessments were blind to subjects’ groups. All patients were subjected to a series of evaluations, including spatial navigation tests, psycho-cognitive assessments, and MRI imaging, which will be elaborated upon in the subsequent sections.

### 2.1. Spatial Navigation Test

#### Virtual Morris Water Maze (vMWM)

The vMWM served as our primary tool for assessing spatial navigation capabilities [51]. The original paradigm was designed for rodents, allowing them to swim freely in a round pool adorned with visual cues. Within the pool lies a transparent platform, hidden slightly underwater from the rodent. To rest, the rodent must first locate and remember this platform’s position, improving its efficiency in reaching it in subsequent trials. Memory impairments, such as from hippocampal lesions, cause rodents to fail in locating the platform. Adapted virtual versions of this test for humans have been validated to identify memory deficits, including those seen in Alzheimer’s patients. [52,53,54].

The vMWM has also been widely used to assess spatial navigation abilities in individuals with vestibular disorders, such as bilateral vestibulopathy [55,56]. Our group previously implemented the vMWM, revealing a pronounced and distinct impairment in spatial navigation skills among PPPD patients [12].

In this study, all tests were conducted 1.5 m apart from a 24.5-inch desktop monitor. Participants navigated a virtual environment using a joystick, facilitated by Simian Labs-Maze Engineers^®^’ Morris Water Maze Software (Build 20210821), working on an MSI GT75 Titan computer with a 9SG Intel i9-9980 processor and an NVIDIA RTX 2080 graphics card, Micro-Star Int’l Co, New Taipei City, Taiwan. This virtual environment comprised a square room (1 × 1 virtual distance units in both “north–south” and “east–west” dimensions) with visual cues centrally positioned on all four walls. At the room’s center was a round pool of 1 virtual unit in diameter.

Considering the risk of virtual-reality-induced motion sickness, particularly in PPPD patients sensitive to visual motion, we preemptively addressed discomfort. Participants were advised to cease the assessment if dizziness occurred and report it. All patients completed the non-immersive task without discomfort.

Our vMWM testing protocol was structured as follows:Block Aconsisted of two training trials: (i) free navigation was allowed for familiarization with joystick movements; (ii) participants were instructed to navigate towards the visual cues on the room’s walls as a practice exercise.Block Bencompassed four consecutive trials. A visible red square platform, measuring 0.17 × 0.17 virtual units was consistently located in the pool’s north-eastern quadrant. Participants initiated each trial at the southern end of the pool, facing north. Successful arrival at the platform was indicated by a rewarding sound, signaling trial completion. This block was designed to familiarize participants with the virtual environment and the test protocol. The maximal trial duration was set to 1 min for this and all subsequent blocks, irrespective of whether the target was reached or not. Blocks A and B serve as initial tests to identify any motor or group-specific issues related to manipulating the joystick and navigating within the virtual environment. We have chosen to include data from these blocks to emphasize that differences observed in subsequent blocks are indeed attributable to variations in navigational skills and not influenced by other potential confounding factors.Block Cincluded seven consecutive trials. Starting from the same southern location, participants were tasked with locating a hidden platform situated in the south-western quadrant. This platform became visible only upon direct contact and emitted the same rewarding sound. Participants were instructed to remember and optimize their route to the platform across the trials. The visual cues provided were consistent: old-style airplanes (north), a sea turtle (west), “La Gioconda” by Leonardo da Vinci (south), and flowers (east). This phase represented the initial setting that necessitated the application of either egocentric or allocentric spatial navigation strategies. Participants were required to memorize the location of the hidden platform and utilize these strategies to efficiently locate it in subsequent trials.Block Dwas a mirrored version of Block C, comprising another set of seven trials. Here, the hidden platform was relocated to the north-western quadrant. The visual cues were changed to simple colored geometric symbols: a black cross (north), a red square (west), a pink heart (south), and a blue triangle (east). Block D closely resembles the preceding block, with the distinction of employing visual cues of markedly lower complexity, devoid of any emotional connotation. These cues consisted of simple geometric symbols as opposed to the more evocative animals or artworks used previously. The aim of this design choice was to evaluate the potential emotional influence exerted by the visual cues on the navigation process.Block Einvolved a sequence of seven trials similar to Blocks C and D. The hidden platform was positioned on the pool’s western side. New visual cues featured a flying condor bird (north), a sunflower field (west), Van Gogh’s “La Méridienne” (south), and an old train (east). This block introduced an increased spatial navigation challenge by incorporating random starting positions and initial facing orientations. Typically, such a “random start” setting is believed to maximize the reliance on allocentric navigational mechanisms [52].Block Fconsisted of four trials, similar in complexity to Block B, with a distinctly visible platform positioned in the south-east quadrant. It employed random starting points, as in Block E, to serve as a control for the assessment of joystick manipulation and movement within the virtual environment.

Various metrics can assess navigational performance in the MWM paradigm, including path length, latency, and time spent in the target’s quadrant. An especially sensitive metric is Gallagher’s proximity, or the cumulative search error (CSE), representing the average distance (measured in virtual units equivalent to 1 Morris water maze pool diameter) between the subject and the target at every timepoint during the trial. This metric highlights the efficiency of the search strategy, indicating whether the subject navigates closer to or further away from the target, even if the hidden platform is not directly located. In our study, the CSE served as the primary metric for quantifying spatial navigation errors in Blocks B through F.

### 2.2. Global Cognition

#### Montreal Cognitive Assessment (MoCA)

The MoCA is an efficient tool for detecting mild cognitive impairment, assessing various domains such as attention, visuospatial skills, executive functions, memory, language, calculation, and orientation. With strong psychometric properties, it is sensitive and specific across diverse groups and is validated by various normative data [57]. We considered it to be a reliable and sensitive indication of overall cognitive dysfunction.

### 2.3. Memory and Attention

#### 2.3.1. Digit Span Task (DST)

The Digit Span Test (DST), part of the Wechsler Adult Intelligence Scale, showing high test–retest reliability and construct validity for assessing attention and short-term memory [58], through 8 items with 2 trials each, requiring subjects to recall increasingly longer digit series. It includes a forward and backward version, the latter also assessing working memory and executive functions [59]. The test concludes after two consecutive failed attempts on an item, with a maximum score of 16 for each version. We selected the DST to evaluate auditory–verbal working memory and attention, components that are less reliant on visuospatial processing.

#### 2.3.2. Corsi Block-Tapping Task (CBTT)

The Corsi Block-Tapping Task (CBTT) assesses visuospatial working memory by having participants replicate sequences tapped by an examiner on a 9-block board, both directly and in reverse. It progressively increases in difficulty and is considered a reliable and valid measure with an age-adjusted normative scores [60,61]. The CBTT is sensitive to visuospatial memory deficits in patients with vestibular pathology [61,62]. Considering that effective spatial navigation is contingent upon robust spatial memory, we deemed it essential to evaluate this cognitive domain independently using the CBTT.

### 2.4. Visuospatial and Executive Functions

#### 2.4.1. Trail Making Test A and B (TMT-A and TMT-B)

The Trail Making Test (TMT) involves connecting 25 numbered circles (TMT-A) or alternating between numbers and letters (TMT-B) on paper, assessing attention, visuospatial scanning, and processing speed, with TMT-B also evaluating executive functions. Completion time is the key metric. Both versions are validated and normed across age groups [63]. We incorporated the Trail Making Test (TMT) in both its versions to assess more complex aspects of spatial processing, acknowledging that the cognitive dysfunction in PPPD may extend beyond spatial memory or navigation alone.

#### 2.4.2. Tower of London Test (ToL)

In the Tower of London (ToL) test, participants arrange colored discs on pegs to match a given pattern, testing executive functions and visuospatial planning skills. The test possesses good internal consistency and construct validity for executive functions [64,65,66]. The difficulty increases over 12 trials, with scoring based on the number of moves to achieve the target arrangement. The test measures accuracy and efficiency, providing an overall “accuracy score” out of a maximum of 36. We chose this test to evaluate what may be considered the most complex of all non-navigational cognitive tasks: the mental imagery and manipulation of spatial objects. This is predicated on the notion that such abstract spatial processing could be indicative of the broader cognitive impairments observed in PPPD.

### 2.5. Anxiety-Depression Assessments

#### 2.5.1. Beck Depression Inventory (BDI)

The BDI is a self-report questionnaire designed to assess the severity of depressive symptoms. It consists of 21 items, each describing a specific symptom of depression. Participants rate how they have felt over the past two weeks on a scale of 0 to 3. Total scores categorize depression severity, ranging from minimal to severe. The BDI is widely recognized for its reliability and validity in both clinical and non-clinical settings [67].

#### 2.5.2. State-Trait Anxiety Inventory (STAI)

The STAI is a commonly used measure for assessing both state and trait anxiety. It comprises two separate 20-item scales. The State Anxiety Scale (STAI-State) evaluates the current state of anxiety, asking individuals how they feel “right now,” while the Trait Anxiety Scale (STAI-Trait) assesses more general and long-standing feelings of anxiety. Participants rate each item on a scale of 1 to 4. Higher scores indicate greater anxiety levels. It is esteemed for its consistency and applicability across diverse populations [68,69].

### 2.6. Vestibular- and Dizziness-Specific Assessments

#### 2.6.1. Dizziness Handicap Inventory (DHI)

The DHI is a self-assessment tool designed to measure the impact of dizziness on daily life. Comprising 25 items, it evaluates the physical, emotional, and functional implications of dizziness-related problems. Respondents indicate “yes”, “sometimes”, or “no” for each question, which correspond to scores of 4, 2, and 0, respectively. A higher cumulative score implies a greater handicap due to dizziness. The DHI is widely used in clinical settings to assess the severity and treatment outcomes of vestibular disorders [70,71].

#### 2.6.2. Analogue Visual Scale for Dizziness (AVSD)

The AVSD is a straightforward tool for patients to rate their dizziness intensity. On a scale from 0 (no dizziness) to 10 (worst imaginable dizziness), patients select a number that best represents their current dizziness severity. This numerical rating aids in assessing the level of discomfort.

#### 2.6.3. Niigata Questionnaire for PPPD (NQ-PPPD)

The NQ-PPPD is a specialized questionnaire developed to assess the severity and characteristics of PPPD. It targets specific symptoms and triggers related to PPPD. Patients rate each item based on their experiences over a given period. A higher score indicates more severe symptoms or functional impairment due to PPPD [5,72].

### 2.7. Vestibular Function

To evaluate vestibular function and ascertain vestibular dysfunction in the test and control groups, video head impulse testing (vHIT) and vestibular evoked myogenic potentials (VEMPs) were administered to all participants.

#### 2.7.1. Video Head Impulse Testing (vHIT)

The vestibulo-ocular reflex (VOR) function of all six semicircular canals was evaluated using the ICS impulse video head impulse testing (vHIT) device by GN Otometrics, Denmark. A total of 20 head impulses were administered per canal. Impulses were deemed valid if they were free of recognizable artifacts and achieved a peak head velocity exceeding 200°/s for the lateral canals and 150°/s for the vertical canals.

#### 2.7.2. Vestibular Evoked Myogenic Potentials (VEMPs)

For recording vestibular evoked myogenic potentials (VEMPs), both ocular (oVEMP) and cervical (cVEMP) responses were obtained using the Eclipse EP25 platform. Standard protocols were followed: oVEMP tests required subjects to sit and gaze upwards at a 30° angle, while cVEMP tests were conducted with the subject supine and the head raised for sternocleidomastoid muscle contraction, monitored for consistent electromyographic signals. A 500 Hz tone burst at 100 dB nHL was used for air-conduction stimulation, with over 100 stimuli per ear averaged to ensure artifact-free, reproducible responses.

### 2.8. Imaging

#### Magnetic Resonance Imaging

Images were acquired at the “Servicio de Resonancia Magnética y Tomografía Computada de la Clínica Alemana de Santiago” using a 3T Siemens SKYRA MRI, Berlin, Germany, system. A neuroradiologist, who was blinded to the clinical evaluations of both volunteers and patients, reviewed and interpreted the images. For this study, the images were specifically examined to rule out additional diseases or any form of structural damage in brain regions crucial for spatial navigation, including the hippocampus.

## 3. Results

Sixty-two patients who met the inclusion criteria were invited to participate. Of these, eleven declined, while fifty-one agreed and completed all assessments. Nineteen patients who met the criteria for PPPD were included in the “PPPD” group. Twenty patients, though not meeting the PPPD criteria, were diagnosed with other vestibular disorders and were placed in the “vestibular” control group. The conditions BPPV, Ménière’s disease, vestibular migraine, and acute vestibulopathy were comparably distributed between the two groups. A separate “control” group consisted of twelve healthy volunteers. The primary characteristics of all participants are summarized in Table 1. Importantly and as intended, no significant difference was found on age (ANOVA F = 0.264; *p* = 0.76) or educational level (ANOVA F = 0.307; *p* = 0.73) between groups (given the known influence of these factors over cognitive performance). The neuroradiological evaluation of magnetic resonance brain scans yielded no abnormalities. Specifically, no hippocampal lesions were identified.

### 3.1. Vestibular Function

To assess group comparability, we rigorously analyzed vestibular function. The PPPD and vestibular groups demonstrated significantly reduced vestibular function compared to healthy controls, as determined by vHIT and VEMP assessments (ANOVA with Tukey post hoc, *p* < 0.05). Despite this reduction, there was no significant difference between the PPPD and vestibular groups, suggesting similar levels of dysfunction. Table A1 in Appendix A contains comprehensive results, including mean VOR gain, the proportion of patients with gains below 0.7, corrective saccades, and VEMP response amplitudes, along with detailed statistical analyses for each variable.

### 3.2. Spatial Navigation

In the vMWM trials, PPPD patients showed significantly reduced spatial navigation abilities compared to the vestibular and control groups (Figure 1). Due to heterogeneous variances (Levene’s Test W = 4.185; *p* = 0.021), non-parametric tests were utilized, where the Kruskal–Wallis results were significant (H = 20.6; *p* < 0.0001). Subsequent post hoc Dunn tests indicated differences between the PPPD and vestibular groups (*p* = 0.014), and between the PPPD group and healthy controls (*p* = 0.0003). However, the vestibular and control groups did not differ significantly (*p* = 0.13).

Analysis of each group’s behavior during specific vMWM settings (Figure 2) showed no noticeable difference during free navigation in training trials (Block A, Kruskal–Wallis W = 1.804; *p* = 0.406). This was also the case when subjects were directed straight towards a visible target, whether from a fixed starting point (Block B, W = 3.95; *p* = 0.166) or a random starting point (Block F, W = 4.47; *p* = 0.107).

However, distinctions became evident when the target remained hidden until located. This presents a genuine spatial navigation challenge, where one must rely on egocentric or allocentric cues, such as a cognitive map, to accurately reach the location where the hidden target is expected to be based on the experience from previous trials [12]. In such cases, the PPPD group performed significantly worse than the vestibular and control groups. This trend was evident in both blocks with a fixed starting point (Block C: Kruskal–Wallis W= 17.5; *p* < 0.001; post hoc Dunn *p* < 0.038; Block D: W= 9.49; *p* = 0.009; post hoc Dunn *p* < 0.04). The difference was even more pronounced when the starting point was randomized (Block E: Kruskal–Wallis W= 22.7; *p* < 0.001; post hoc Dunn *p* < 0.0002). The comparison of effect sizes, measured by Cohen’s d, between blocks showed that the magnitude of difference in Block E (d = 1.32) was significantly larger (Cohen’s Q test = 6.25; *p* = 0.044) than in Block C (d = 0.82) and Block D (d = 0.45).

Assessing each block’s spatial learning progress is possible by observing the decrease in CSE scores after sequential trials. Figure 3 shows that all groups improved across trials. However, the learning curve was steeper for the vestibular and control groups. In contrast, the PPPD group’s learning pace was slower, and did not match the performance levels of the other two groups within the seven trials. Statistical analysis via the Friedman test (*p* < 0.001) confirmed this, with Bonferroni post hoc testing substantiating the differences between the PPPD group and the other two groups.

Figure 4 (upper row) presents heatmaps illustrating navigation during Block C. The healthy controls primarily remained near the target, whereas the vestibular patients exhibited broader movement within the pool. The PPPD patients showed the greatest navigational spread, confirmed by a higher standard distance deviation, indicating significantly more dispersion compared to the vestibular and control groups (Levene’s W = 232.1; *p* < 0.001).

Figure 4 (lower panel) illustrates the navigational routes of a single subject from each group during Block D of the vMWM, with the seven trials represented in different shades of blue. The paths show that both the vestibular and healthy control groups, after initially finding the target within the first few trials, could navigate back using direct routes. In contrast, PPPD patients showed erratic patterns, suggesting difficulties with spatial memory retention or strategic navigation. Notably, PPPD subjects often stuck to the pool’s periphery and sometimes moved in tight, localized circles, possibly indicating a hesitance to explore the environment fully.

### 3.3. Non-Spatial Navigation Cognitive Tests

Figure 5 details the performance of each group on cognitive tests not related to spatial navigation. Variance homogeneity was verified (highest Levene’s W = 2.27, *p* = 0.115), permitting ANOVA analysis for group differences, with post hoc Tukey tests clarifying disparities. For clarity, certain test scores were inverted (MoCA, DST, CBTT, and ToL), so that larger values uniformly indicate more severe impairment. PPPD patients demonstrated greater deficits on the Niigata (PPPD symptoms), DHI (dizziness impact), AVSD (dizziness severity), STAI-Trait (long-term anxiety tendencies), MoCA (global cognition), TMT-B (executive function in visual-spatial planning), CBTT (spatial memory), and ToL (executive function in visuospatial planning) compared to vestibular and healthy controls. There were no observed differences in STAI-State (anxiety at the moment), DST (digit retention memory), or TMT-A (processing speed on a visuospatial task). Vestibular patients exhibited higher BDI scores (depressive symptoms) than controls.

### 3.4. Correlations between Cognitive Tests

Furthermore, beyond comparing this difference between groups, and recognizing how every cognitive test in our study (including spatial navigation performance) depends on many different factors, an exploratory correlation analysis was carried out, followed by a more in-depth factor analysis. Figure 6 shows the Pearson correlation coefficient for each pair of variables presenting a correlation associating a *p*-value lower than 0.05. Focusing primarily on our key variables, Niigata (indicative of PPPD symptom severity) and CSE (a measure of spatial navigation impairment), several correlations emerged.

Niigata (PPPD symptomatology) correlated significantly with DHI (dizziness impact on daily life), BDI (depressive symptomatology), DST (working memory), and CBTT (spatial memory). CSE (navigation performance) correlated significantly with Niigata, DHI, MOCA (overall cognition), WAIS (working memory), TMT_A (processing speed for a visuospatial task), Corsi (spatial memory), London Towers (executive functions on a visuospatial task), and age.

### 3.5. Factor Analysis

To explore potential underlying constructs that could explain the interconnectedness of our cognitive tests, we employed a factor analysis with varimax rotation, and retaining all factors with an eigenvalue larger than 1. Three distinct factors emerged. We present the weight of each of our variables on each factor as a heatmap in Figure 7. Given the distribution of variable weight across factors and choosing the main variables of a factor as those with high weight on one factor and lower or opposite weight on others, we suggest the following naming and interpretation for our emerging underlying factors:

#### 3.5.1. Factor 1: PPPD Severity and Its Impact on Daily Life

Within this factor, several variables co-vary in the same direction, specifically, Niigata (PPPD symptoms), DHI (dizziness impact), AVSD (dizziness severity), BDI (depressive symptoms), STAI-Trait (long-term anxiety tendencies), and CSE (spatial navigation errors). This factor seems to encapsulate the “core” manifestation of PPPD symptoms in subjects. Niigata, DHI, and AVSD hold the greatest weight, indicative of the presence of symptoms and their subsequent impact on daily living. The BDI’s prominence suggests that depressive symptoms are influenced by high scores in the DHI and AVSD. The inclusion of trait anxiety (STAI-Trait) but not state anxiety (STAI-State) aligns with the current literature, which posits that trait anxiety acts as a catalyst for PPPD development. Notably, even with a modest weight, spatial navigation impairment (CSE) emerges as a significant variable intertwined with this “core” factor of PPPD symptomatology.

#### 3.5.2. Factor 2: Age-Related Cognitive Changes

This factor seems to function independently from factor 1, enveloping age, educational level, MoCA (global cognition), TMT-A and -B (processing speed and executive function for a visuospatial task), CBTT (spatial memory), and CSE (spatial navigation error). Given the established correlation between age and educational level on various cognitive processes, including spatial navigation, this pattern was relatively expected. Thus, we posit that this factor underscores a dimension distinct from PPPD symptomatology but is heavily influenced by age and educational levels, acting as a “common” factor shaping cognitive performance.

#### 3.5.3. Factor 3: Advanced Cognitive Functions

In this factor, when one variable deteriorates, others follow suit, as seen with DST (non-spatial memory), ToL (visuospatial planning), TMT-B (executive function on visuospatial task), CBTT inverted (more demanding spatial memory), MoCA (global cognition), CSE (spatial navigation error), and Niigata (PPPD symptoms). These selected tests epitomize some of the most challenging, cognitively demanding, or intricate cognitive evaluations. They encompass an array of cognitive processes, ranging from planning and executive functions to visuospatial task assessments. Hence, this factor likely embodies the orchestration or performance of advanced cognitive operations. Intriguingly, while the Niigata score holds some weight within this factor, its association is not as strong as with DHI and AVSD.

## 4. Discussion

Understanding the mechanisms behind functional neurological disorders remains a significant challenge in the field of neuroscience. In this study, we focused on various cognitive tests to evaluate a cohort of patients with PPPD, comparing their results to two control groups: healthy volunteers and individuals with vestibular disorders who did not exhibit PPPD symptoms. Several key insights emerged from our findings:

### 4.1. Impact of Non-PPPD Vestibular Disorders on Visuospatial Memory and Navigation Performance

Research indicates that vestibular loss, especially in bilateral vestibulopathy, can affect visuospatial cognitive functions, including spatial navigation [73,74,75]. It is crucial to distinguish between virtual, non-immersive spatial navigation assessments and real-world navigation due to their differing sensory demands—real-world navigation involves more complex vestibular input, while virtual tasks are more visually based [76,77]. Studies show vestibular loss leads to poorer performance in both real and simulated navigational tasks; however, this may not align with self-reported navigation abilities, suggesting virtual task deficits might not fully translate to real-world navigational challenges [56,76,78].

Our findings show that non-PPPD vestibular loss patients had increased navigational dispersion, shown by a larger standard distance deviation, which was significant. While their navigation patterns resembled healthy subjects’, they covered a broader area en route to the target, complementing the observed trend in CSE metrics and highlighting differential task performance, albeit not reaching statistical significance. The impairment observed was less severe than reported in previous studies, including our own from 2020 [12,55,56,76]. For non-navigational visuospatial tasks, non-PPPD vestibular patients performed worse than healthy controls on the CBTT spatial memory test, consistent with prior findings [62].

### 4.2. Impact of PPPD on Higher-Demand Cognitive Tests

In higher-demand cognitive tests, PPPD patients’ performance was similar to their non-PPPD vestibular counterparts in the CBTT, indicating that visuospatial working memory deficits may stem from vestibular loss rather than PPPD alone. However, PPPD patients fared significantly worse on the MoCA (global cognition), TMT-B (executive functions in the visuospatial domain), ToL (visuospatial planning), and spatial navigation measured by CSE during the vMWM. Despite this, all groups showed comparable performance on the DST for non-spatial working memory and the TMT-A for more basic visuospatial skills. These patterns reveal that PPPD’s cognitive impact is more pronounced in complex tasks requiring advanced visuospatial processing and executive functions, with the disparity amplifying in tasks with escalating cognitive demands.

### 4.3. “Core” PPPD Phenomena

A factor analysis identified a primary factor relating to core PPPD symptoms, significantly comprising the Niigata PPPD score, DHI impact, and AVSD-EVA severity (Figure 7), with these metrics being notably higher in PPPD patients, suggesting a greater perceived disease severity (Figure 5). Understandably, this heightened severity is correlated with increased BDI-depressive symptomatology, likely as a secondary effect of the disease’s burden.

Another noteworthy point in our data is that PPPD patients demonstrated higher levels of trait anxiety (as measured by STAI-Trait) but not elevated state anxiety (STAI-State) when compared to the control groups (Figure 5). Moreover, trait anxiety emerged in factor 1 as a core variable for PPPD, reinforcing the growing view that this trait acts as a predisposing factor for developing PPPD.

While many of the relationships between variables pertain to PPPD severity, impact on life, and known predisposing factors, most of our assessed cognitive variables did not emerge as primary components in this underlying construct. The notable exception is spatial navigation impairment, as gauged by the CSE metric in the vMWM. This cognitive function appears to be intricately linked to the fundamental changes associated with the presence and severity of PPPD in subjects.

### 4.4. Spatial Navigation Is Distinctively Impaired in PPPD

We have successfully replicated our 2020 findings in a new cohort of patients, bolstering the evidence that spatial navigation impairments are a consistent, fundamental feature of PPPD. Despite the inclusion of new controls, updated software, and diverse vMWM setups, PPPD patients consistently showed similar spatial navigation deficits as observed in 2020. Notably, all groups managed comparable performances in Block F, where a visible platform was introduced with randomized starts, suggesting that the PPPD group’s issues are not due to movement but rather arise when tasked with locating a hidden target through cognitive mapping.

Further analysis excluded anxiety or acute discomfort as influencing factors, supported by uniform STAI-State results and the lack of increased dizziness reports during navigation. The data also differentiated between allocentric and egocentric navigation strategies. With randomized starting points in Block E, the reliance on allocentric navigation became essential, and it is here that PPPD patients notably struggled, indicating difficulties in constructing or using cognitive maps, unlike in tasks with fixed starting points.

On a qualitative note, we also found familiar patterns as in our 2020 study: PPPD patients tended to (1) stick close to the pool walls, even when it was clear the hidden target is more centrally located, and (2) move in narrow circles without meaningfully exploring the maze.

### 4.5. Age and Educational Level’s Impact on Spatial Navigation and Cognitive Performance

The factor analysis highlighted that cognitive test performance is influenced by age and educational level, which, while matched across groups and, therefore, not prominent in the rest of our findings, independently affect cognitive variables (factor 2—Figure 7). This underscores cognitive function’s complexity and the importance of considering such factors in PPPD patient assessment, especially in older populations.

### 4.6. Advance Cognitive Functions

Factor 3 from our factor analysis interestingly groups higher-order cognitive functions like TMT-B (visuospatial executive function), ToL (visuospatial planning), and spatial navigation CSE, and is further associated with more complex tasks like the reversed DST (non-spatial memory), reversed CBTT (spatial memory), and MoCA (global cognition), suggesting a trend in PPPD symptomatology. This distinct factor is separate from factor 1’s core PPPD symptoms. Although limited by sample size, these findings hint at a broad cognitive impairment affecting both spatial and non-spatial domains in PPPD patients, a hypothesis requiring validation through larger-scale studies.

### 4.7. Interplay between PPPD Symptomatology and Cognitive Navigational Dysfunction

The cognitive outcomes of PPPD patients, particularly on high-demand cognitive tests, suggest a complex interaction between vestibular dysfunction and PPPD symptomatology. The results from the CBTT indicate PPPD patients’ working memory deficits in visuospatial tasks are comparable to those with non-PPPD vestibular loss, pointing to a general vestibular issue. However, on tasks that require more intensive cognitive engagement, such as the MoCA, TMT-B, ToL, and spatial navigation measured by the CSE in the vMWM, PPPD patients underperform significantly.

Despite similar vestibular functions, the PPPD group experiences more severe difficulties with complex cognitive tasks, indicating that the condition’s characteristic symptoms, like enduring dizziness and balance issues, may be closely linked to—or even worsened by—the cognitive and navigational deficits inherent to PPPD. The data show all groups perform similarly on the DST, which measures simpler cognitive functions, but PPPD patients struggle with higher-order cognitive processes.

This pattern supports the notion that PPPD may amplify vestibular-related dizziness through its effect on cognitive functions vital for navigation, suggesting that PPPD has a unique cognitive signature separate from other vestibular disorders. This calls for an expanded interpretation of PPPD’s impact, recognizing cognitive dysfunction as a crucial aspect of its symptomatology and prompting a reassessment of PPPD’s extensive effects on patient well-being.

### 4.8. PPPD Shows Disruption in Higher-Order Cognitive Functions, Particularly Spatial Navigation: Possible Impact on Cognitive Maps

Drawing from our findings, we suggest that PPPD affects various cognitive functions, particularly in complex tasks involving visuospatial processing, planning, and executive functions, which correlates with the spatial navigation deficits at the core of PPPD. This supports models positing that PPPD involves difficulty in creating risk-assessing vestibular spatial-temporal maps. Assessing the integrity or even the construction of these cognitive maps is both conceptually and technically challenging [28,35,46]. Nevertheless, we interpret our collective data as supporting the disruption of this map-making process. Key points supporting this concept include:Challenges in visuospatial planning and execution in demanding tasks like the TMT-B and ToL, while simpler cognitive functions like memory (assessed by DST and CBTT) remain intact.Issues with creating and using allocentric spatial maps, as seen in the vMWM paradigm.A link between spatial navigation impairment severity and PPPD, suggesting a correlation with higher cognitive functions, including spatial navigation, which could be more pronounced in a subset of patients, according to our factor analysis.

These results could potentially be confirmed by neuroimaging studies like EEG or fMRI, which could validate disruptions in perceptual mapping as central to PPPD. Speculatively, cognitive training in spatial navigation might reverse neural changes in PPPD, presenting a new therapeutic strategy for severe cases unresponsive to existing treatments.

The preliminary data indicate spatial navigation tests could be valuable in PPPD diagnosis, although this is speculative given our study’s limitations. Since PPPD diagnosis currently relies on subjective criteria, further research with larger samples and robust methods is needed to verify these tests’ diagnostic value.

## 5. Limitations

Our study’s conclusions are bound by certain constraints, including a modest sample size of 19 PPPD patients, which nevertheless aligns with methodological guidelines suggested by some authors, who recommend a minimum of 3–6 subjects per variable in factor analysis [79]. Each identified factor had an eigenvalue over one, suggesting relevant variance explanation.

We assessed the potential influence of comorbid symptoms like dizziness and anxiety on cognitive performance and found no significant anxiety differences as per the STAI-State test, indicating spatial dysfunction might not be anxiety-driven. Yet, we cannot discount the role of overall PPPD symptomatology in these dysfunctions.

The study’s multiple cognitive tasks necessitate consideration of potential order effects, like practice or fatigue, on performance. We managed these by escalating task complexity and were mindful of fatigue, particularly in Block F, designed to minimize the need for spatial navigation. Future studies may benefit from counterbalancing the task order to mitigate such effects.

While we propose spatial navigation impairments as characteristic of PPPD, we acknowledge the limitation posed by the small sample and PPPD’s heterogeneity. The robustness of these findings must be evaluated in larger, more diverse cohorts to enable generalization across the broader PPPD population.

## 6. Conclusions

In our study, we have presented behavioral data from various cognitive tests, suggesting that PPPD may be associated with neural shifts in brain networks that are involved in high-demand cognitive functions. Spatial navigation abilities seem particularly affected and appear to be a central feature of the disease.

We, as authors, put forth the notion that these findings could support the idea that PPPD involves a disruption in the creation of context-dependent, risk-integrating perceptual maps. This is an interpretative lens that aligns with some aspects of the existing literature [17,28,33] but is, of course, subject to further validation.

Future research should ideally include electrophysiological and functional imaging metrics during tasks involving high-level vestibular and spatial navigation functions, as these could provide more conclusive evidence to support or challenge our hypotheses.

The impairment of spatial navigation skills and other cognitive function would support the notion of PPPD being a broader disorganization of multiple higher-order cortical functions, which would not only be related to vestibular processing but, more significantly, are essential in constructing a reliable internal perceptual map of the external world and the body’s position within it. In our proposal, these broad cognitive dysfunctions would affect the reconciliation between predicted perceptual maps (filled with errors) and the re-afferences of the actual external world, placing this at the core of PPPD phenomena.

## Figures and Tables

**Figure 1 brainsci-14-00016-f001:**
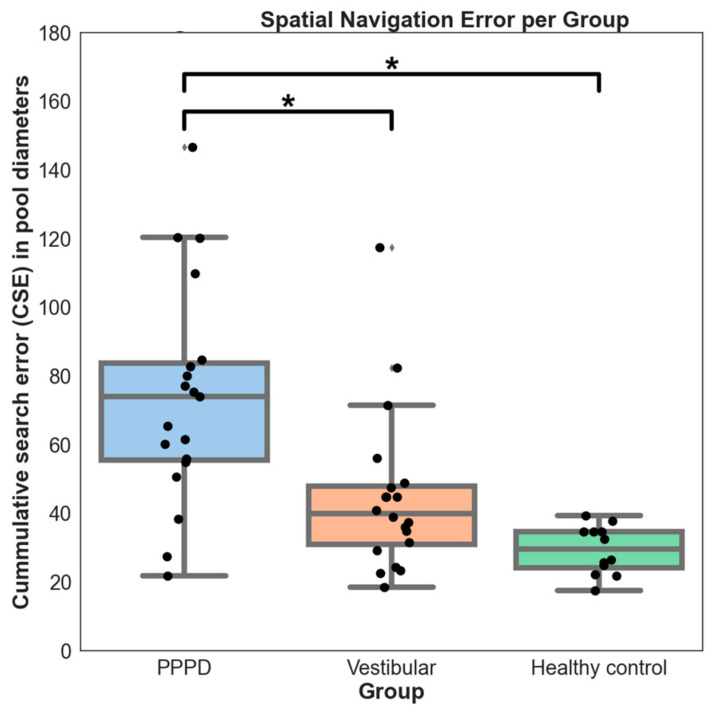
Spatial navigation error per group. Cumulative search error (CSE) describes performance during the virtual Morris water maze (vMWM), thus representing the amount of impairment of spatial navigation skills. When considering all trials of vMWM, the PPPD group showed significantly worse performance than vestibular and control groups (marked ***** for *p*-values < 0.014). There was no difference between vestibular and control groups. Black circles represent individual subjects.

**Figure 2 brainsci-14-00016-f002:**
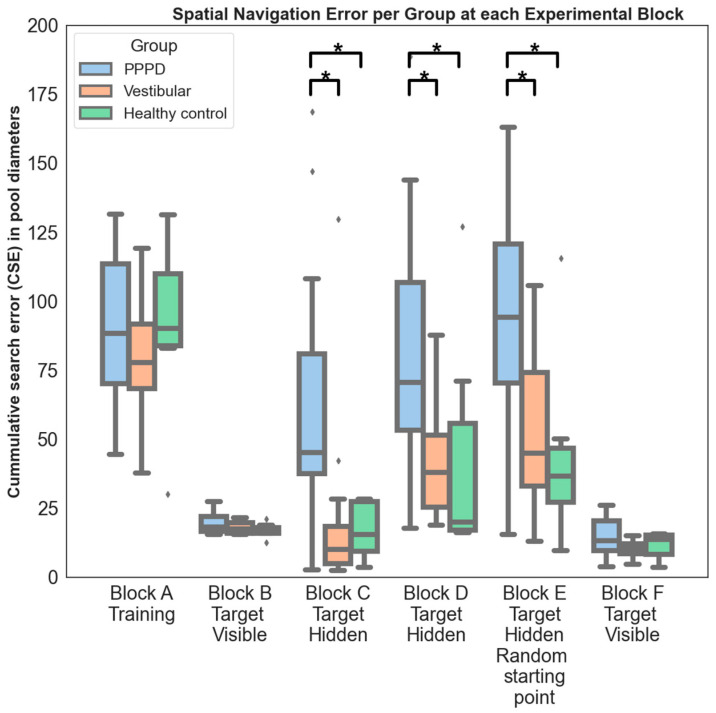
Spatial navigation error per group on three different settings (blocks) for vWMW. Same metric as in Figure 1 but separated on every setting of the virtual Morris water maze (vMWM) in our experiment (* = significant difference with *p*-value < 0.05; see text for more detailed description). **Block A—training**: no difference between groups (no target present, CSE was calculated in relation to center of pool). **Block B—target visible/fixed starting point:** no spatial navigation challenge. No difference between groups. **Block C—target invisible/fixed starting point**: spatial navigation challenged. PPPD showed worse performance than vestibular and control groups. **Block D—target invisible/new set of visual cues/fixed starting point**: spatial navigation challenged. PPPD showed worse performance than vestibular and control groups. **Block E—target invisible/new set of visual cues/random starting point**: most challenging setting for spatial navigation in our experiment. PPPD showed worse performance than vestibular and control groups, with this difference being larger than in previous settings (see text for detailed statistics). **Block F—visible target/random starting point**: no spatial navigation challenge. No difference between groups.

**Figure 3 brainsci-14-00016-f003:**
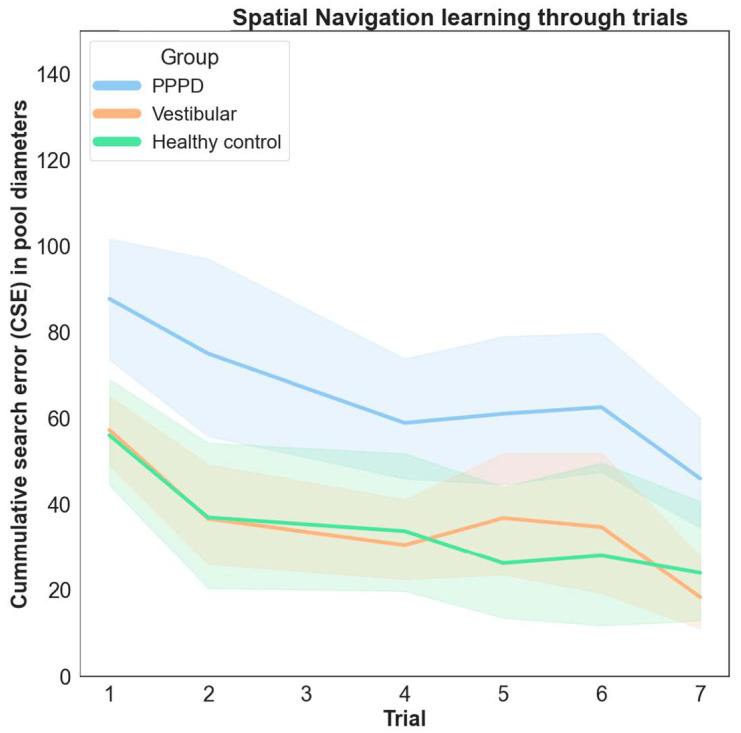
Spatial learning through vMWM trials. CSE scores averaged together across Blocks C–E (all those which where navigationally challenging) at each trial from first to seventh (last trial). The 95% confidence intervals are shown by the colored areas. The decrease in CSE towards later trials represents increasingly better performance after each trial. The “learning slope” is very similar between vestibular and control groups, reaching almost optimal navigation during the 4 first trials. The PPPD group showed worse performance through the learning trials.

**Figure 4 brainsci-14-00016-f004:**
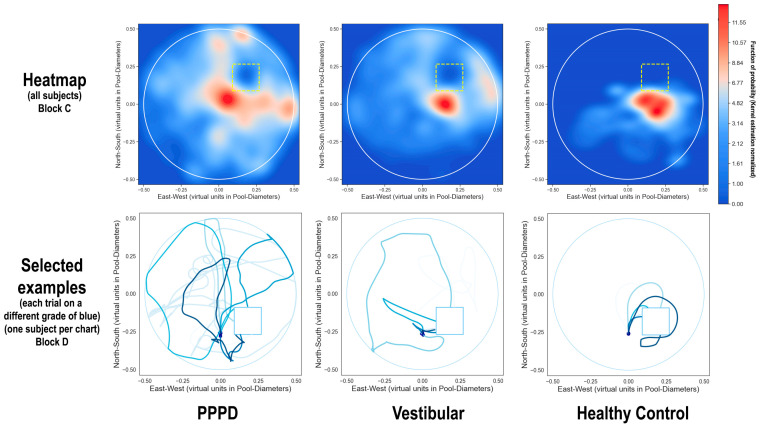
Navigation patterns in vMWM pool. The upper row showcases density plots derived from kernel estimations, highlighting frequently visited areas during Block C navigation, which illustrate each group’s overall behavior in relation to the proximity of the hidden target, indicated by yellow dotted squares. Red regions denote the most frequented areas, while blue indicates lesser visited zones, normalized across 50 probability levels. Healthy controls predominantly navigated near the target, indicated by prevalent deep blue areas, suggesting a low likelihood of finding a control subject elsewhere. Conversely, vestibular patients displayed a broader, yet still target-focused navigation pattern. PPPD patients, while primarily directed towards the target, traversed a more expansive range of areas, with certain regions, particularly near the pool’s walls, showing a higher likelihood of PPPD patient presence. The lower row highlights individual navigation paths from selected cases of each group in Block D, with varied shades of blue representing each trial’s trajectory.

**Figure 5 brainsci-14-00016-f005:**
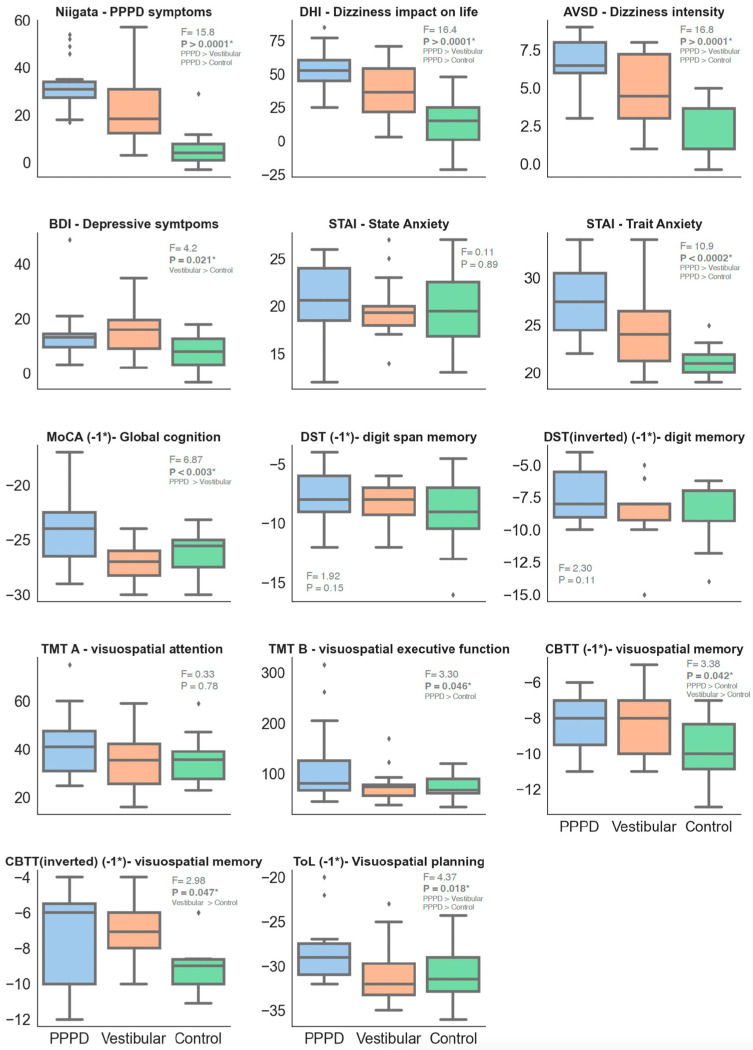
Results from neurocognitive tests. To facilitate interpretation, some test values were inverted (−1*), ensuring that in every subplot, larger values indicate greater impairment or worse performance. ANOVA results are presented for each test. When a significant difference emerged, a post hoc Tukey test was conducted. Specific paired differences are noted when the *p*-value is less than 0.05.

**Figure 6 brainsci-14-00016-f006:**
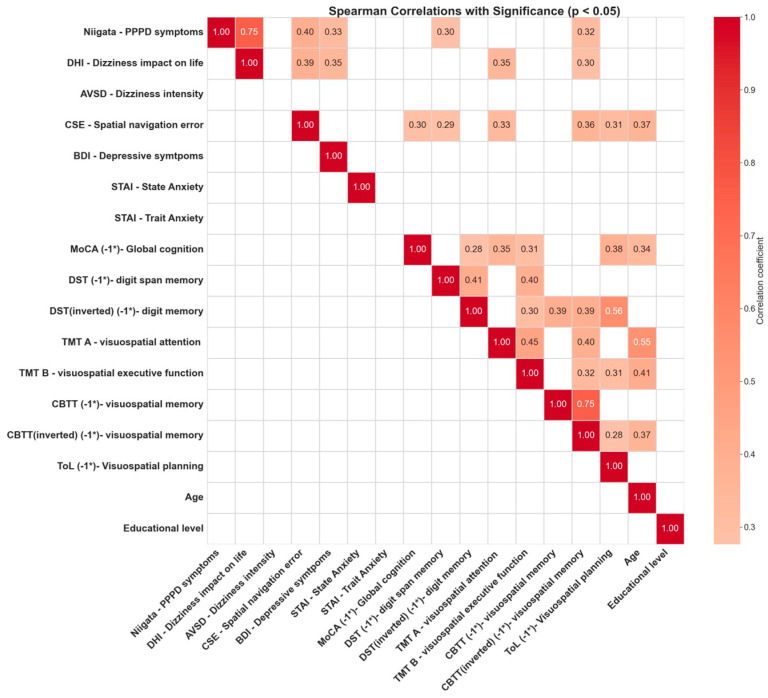
Significant correlation between spatial navigation performance neurocognitive tests. Pearson correlations were explored between all major variables. To facilitate interpretation, some test values were inverted (−1*), ensuring that for all tests, larger values indicate greater impairment or worse performance. Only correlations associated with a significant *p*-value (*p* < 0.05) are shown.

**Figure 7 brainsci-14-00016-f007:**
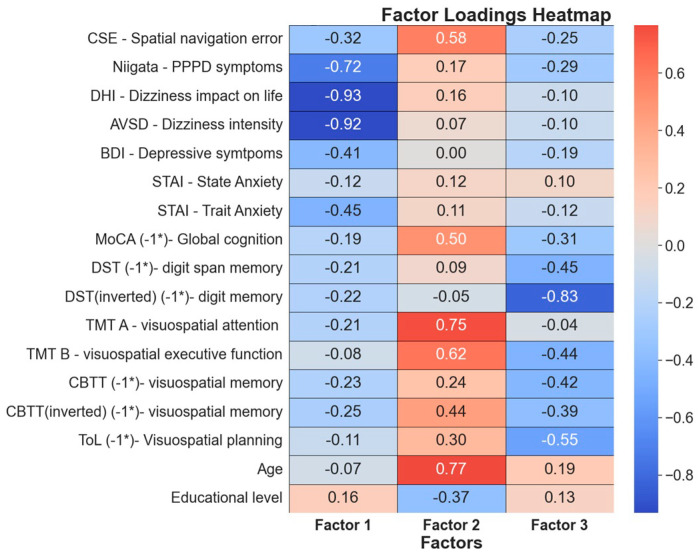
**Factor analysis.** Three underlying factor were selected with eigenvalues larger than 1. The weight on each factor of each variable in the study are presented in this heatmap table. To facilitate interpretation, some test values are inverted (−1*), ensuring that for all tests, larger values indicate greater impairment or worse performance. We have chosen to name these factors **factor 1—PPPD severity and associated distress** (this factor includes Niigata, DHI, AVSD, STAI-Trait, BDI, and CSE); **factor 2—age related cognitive changes** (including age, educational level, MoCA, TMT-A and -B, CBTT, CSE); and **factor 3—advance cognitive functions** (including DBT, London, TMT_B, Corsi inverted, MOCA, CSE, Niigata).

**Table 1 brainsci-14-00016-t001:** Demographic summary of PPPD, vestibular, and control groups.

		Group		
		PPPD	Vestibular (Non-PPPD)	Healthy Control
Number		19	20	12
Age *	Mean	46.8	44	43.6
	Standard deviation	14.9	13.9	15.3
	Range	21–65	20–63	25–64
Gender	Female/Male	79%/21%	85%/15%	75%/25%
Educational level *,^†^	Mean score	3.78	3.9	3.75
	Standard deviation	0.53	0.31	0.46
Diagnosis (percentage of each group)	Vestibular migraine	21.6%	33.3%	-
	Vestibular neuritis	24.3%	23.3%	-
	Benign positional paroxysmal vertigo	10.8%	23.3%	-
	Bilateral vestibulopathy	2.7%	10%	-
	Otoesclerosis	2.7%	3.3%	-
	Meniere’s disease	2.7%	3.3%	-

* No significant difference was found on age (ANOVA F = 0.264; *p* = 0.76) or educational level (ANOVA F = 0.307; *p* = 0.73) between groups. ^†^ Educational level was classified as 1 = primary education incomplete, 2 = primary education complete, 3 = secondary education complete, 4 = undergraduate education complete, 5 = postgraduate education complete.

## Data Availability

All original data can be found at: https://www.labonce.cl/projects-6 (accessed on 11 October 2023).

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
