# Peer review of "Functional Dizziness as a Spatial Cognitive Dysfunction"

_brainsci, 2023, doi:10.3390/brainsci14010016_

Round 1
Reviewer 1 Report
Comments and Suggestions for Authors
Functional Dizziness as a spatial-cognitive dysfunction
The authors report a basic science experiment in which they compared spatial and non-spatial cognitive functions across three groups, PPPD, non-PPPD vestibular dysfunction, and healthy controls. This study is a follow-up to the author’s 2020 paper with similar aims. The attempt to address issues experienced by a challenging patient population, for which we have limited understanding of their pathophysiology and for whom limited interventions are available. At this point, there are numerous issues with the manuscript that are outlined below and that must be resolved before it is suitable for publication. Chief among these is that the authors have not described or apparently controlled for underlying vestibular dysfunction in any of the groups. Related is that the PPPD group was far more symptomatic than the vestibular group. Thus, this reviewer is not convinced that the authors data supports their stated conclusions.
Abstract
Line 17: When written out, PPPD should not be capitalized. The abstract is light on the details of the methods and this should be bolstered. Additionally, it is not clear how the final sentence is supported by the rest of the abstract, please revise.
Introduction
Line 38: Here, and throughout the manuscript, a space is missing between the text and the citation. Please check the style guide and correct errors.
Line 43: The use of “PPPD’s epidemiology…” is awkward, please revise here and elsewhere.
Line 51: It is not clear what is meant by “neural changes,” particularly given that the preceding text describes a lack of lesions in the CNS. Please clarify.
Line 52: Previously the authors state that the pathophysiology of PPPD is not fully understood; yet, here they make rather definitive statements about the causation of symptoms of PPPD. The fact is that the genesis of these signs and symptoms is not known. Please revise this paragraph.
Line 58: What is meant by a “neural shift?”
Line 62: Revise to eliminate passive voice.
Line 65: This sentence is difficult to read and has several grammatical errors, please revise.
Line 68: This sentence is also awkwardly constructed and needs revision. Further, what is meant by “while some trends recognize…”? The phrase “opting instead” implies the vestibular inputs themselves chose their path, please revise. Further, although vestibular afference is integrated with other sensory afference at all levels of the brain, the authors should discuss the so-called vestibular cortex.
Line 80: The next three paragraphs are not paragraphs, but rather single sentences. Please revise to form a complete paragraph and use active voice.
Line 95: Add a space between (i) and patients. Also correct the use of capitalization and punctuation in this sentence.
Line 97: Although the authors found between-group differences, the inclusion of vestibular migraine in the non-PPPD group is problematic because this condition often coexists with PPPD or these two conditions can be challenging to differentially diagnose. Thus, this could contaminate the non-PPPD group. This could diminish any between-group differences. Please address this concern.
Line 102: The also report the inclusion of those with “acute vestibulopathy,” but then provide characteristics for these individuals that imply they were in a sub-acute or chronic phase of recovery. Please clarify.
Line 120: Although the vMWM task has been implemented in persons with vestibular dysfunction, evidence demonstrates that performance, and the types of errors seen, on the vMWM task differ from that of real-world spatial navigation tasks. The differences between virtual and real-world navigation and the implications these differences have on understanding the effects of vestibular dysfunction on spatial navigation and the interpretation of these results must be thoroughly discussed in the manuscript. Additionally, the specific details of the computer hardware used must be provided. Further, since PPPD patients are often sensitive to visual motion, the authors must report how they monitored and managed symptoms of dizziness during the testing.
Line 142-169: Revise so complete sentences are used throughout. Also, it seems problematic that the authors changed multiple variables across the block, e.g., visual cues, start location, and whether start direction were randomized. Please justify the choices made. Also, please discuss the potential order effects related to asking the subjects to perform these blocks in the same sequential order.
Line 170: The choice of CSE as the primary metric for analyzing performance on the vMWM task is appropriate; however, the other metrics have value too and may shed further light on the challenge. Findings from these other metrics could be shared in an appendix.
Line 178: Table 1 does not add significant value; this information should be included in the text. Also, for each of the cognitive assessments, the authors should provide details regarding the psychometric properties. Further, please justify the inclusion of each test, why did you think these particular tests would be useful?
Line 257: These are commonly referred to as visual analogue scales (VAS). Please revise.
Line 265: PPPD does not need to be spelled out again. Is there a reference to support its sensitivity?
Line 288: Unless the authors conducted statistical tests to determine between-group differences in demographic factors, it is not appropriate to state “no significant difference was found on age…” Either report the statistical test findings or revise this statement.
Line 303: The inclusion of results for Blocks A and B does not add value to the manuscript. Why would did the authors think there might be differences in these block? This could be removed from the text and figures, which improve allow for less cluttered presentation of the results.
Figure 2: These data suggest that was considerable within group variability in performance. Did the authors test for homogeneity of variance? What were those results? Are parametric statistics appropriate?
Line 327: What test was used to compare effect sizes? Please report.
Line 332: Better to state that the performance of the PPPD group did not reach the level of the other groups within seven trials. Figure 3 suggests that the PPPD group may have reached an equivalent level of performance within 10 or so trials.
Line 365: Some of this paragraph is Discussion. Please revise to ensure the text within each section is appropriate to that section.
Line 382: This sentence is not necessary. Please delete.
Line 398: The next two sentences are not paragraphs. Please revise. Actually, this is a problem throughout the manuscript. There are several places where only one sentence is devoted to a paragraph. This needs to be corrected throughout.
Line 423: The presentation of the regression data is incomplete, which makes it difficult to evaluate. With such few data point, Figure 7 is also not convincing. The co-efficients, SE, etc are needed. What type of modeling was used? Did the authors control for any factors? Wei et al 2020 have already suggested that cognitive factors influence spatial navigation. Did the authors control for cognition? It is well known that vestibular function affects spatial navigation. Did the authors control for residual vestibular function? It seems unlikely since vestibular function test results are not mentioned anywhere in the manuscript. This is a significant confounding issue, and must be reported appropriately.
Line 493: The content is this section is a mix of visual-spatial cognition and spatial navigation. So, it doesn’t all seem to fit under the current heading.
Line 502: It is not clear how this statement jibes with the significant results for CSE presented in tables 1-3. Please revise.
Line 509: Given the significant between-group differences in Niigata (which you would expect), DHI and AVSD, this reviewer finds it difficult to conclude that PPPD is what has an impact on higher cognitive functions… and spatial navigation for that matter. The vestibular comparison group was much less symptomatic. One cannot be certain that PPPD is the issue, rather it could just be that these people with PPPD were more symptomatic than these people with other vestibular conditions. The level of symptoms could be the issue affecting cognition and spatial navigation. This is particularly difficult to sort out since we don’t know anything about the vestibular function of the enrolled participants, as mentioned earlier. If there are more people with actual vestibular hypofunction in the PPPD group, that could skew these results. This was an issue in the author’s 2020 paper as well.
601: This limitation cannot be understated. It is difficult to place much weight on the factor analyses as presented.
Line 608: Again, the strength of these conclusions is diminished by the lack of information about vestibular function and the strong correlations between symptoms/severity and performance. An equally symptomatic vestibular group might have fared just as poorly.
Line 637: This is too big of a leap considering the limitations of this study.
Comments on the Quality of English Language
There are minor issues, such as capitalizations, plural forms, sentence and paragraph structure.
Author Response
Dear Reviewer,
We would like to extend our deepest gratitude for the time and effort you have dedicated to reviewing our manuscript. Your insightful comments and suggestions have been invaluable, and we have taken careful steps to address each point you raised. We believe that these revisions have significantly enhanced the quality and clarity of our work, and we are pleased to present a new version of the manuscript that reflects these improvements.
Enclosed with this correspondence, you will find the revised manuscript, alongside a Word document that outlines your initial comments. In this document, we have provided our responses in green text following each comment to facilitate an easy review of the changes made and the rationale behind them.
We appreciate the opportunity to refine our manuscript with your guidance and look forward to any further suggestions you may have.
Review answers.
Reviewer 1:
Functional Dizziness as a spatial-cognitive dysfunction
The authors report a basic science experiment in which they compared spatial and non-spatial cognitive functions across three groups, PPPD, non-PPPD vestibular dysfunction, and healthy controls. This study is a follow-up to the author’s 2020 paper with similar aims. The attempt to address issues experienced by a challenging patient population, for which we have limited understanding of their pathophysiology and for whom limited interventions are available. At this point, there are numerous issues with the manuscript that are outlined below and that must be resolved before it is suitable for publication. Chief among these is that the authors have not described or apparently controlled for underlying vestibular dysfunction in any of the groups. Related is that the PPPD group was far more symptomatic than the vestibular group. Thus, this reviewer is not convinced that the authors data supports their stated conclusions.
Author’s Response: Thank you for your insightful review and the issues you have raised regarding our manuscript. We appreciate the opportunity to clarify the points concerning the vestibular assessments conducted within our study groups.
Upon reflection, we recognize that our decision to omit detailed vestibular function data from the manuscript inadvertently led to a lack of clarity in demonstrating the comparability between our PPPD group and the non-PPPD vestibular dysfunction group. This was an oversight on our part. Initially, we were concerned that including extensive raw data might overwhelm the manuscript and detract from its focus. However, we now understand the critical nature of this information in substantiating our conclusions.
In light of your feedback, we have revised the manuscript to include a comprehensive account of the vestibular function tests performed. This additional data unequivocally supports the comparability of the vestibular function between the PPPD and non-PPPD vestibular dysfunction groups, which is a pivotal aspect of our study's findings.
We are confident that this amendment will address the concerns you have highlighted and reinforce the validity of our conclusions. The revised section now provides a clear and detailed presentation of the vestibular assessments, ensuring that the manuscript accurately reflects the rigor of our research methods.
Once again, we are grateful for your valuable critique, which has undeniably strengthened our manuscript. We look forward to your re-evaluation.
Changes: New Section in methodology lines 356 – 395
New Table 2
New Results lines 422 to 44 – 3.1 Vestibular function
Abstract
Line 17: When written out, PPPD should not be capitalized. The abstract is light on the details of the methods and this should be bolstered. Additionally, it is not clear how the final sentence is supported by the rest of the abstract, please revise.
Author’s Response: Thank you for your valuable feedback. We have addressed both points in the revised abstract: the term PPPD is now correctly uncapitalized when written out, and we have expanded the methods section for clarity. Additionally, we have revised the concluding sentence to ensure it is well-supported by the preceding content.
Complete new abstract lines 19 to 34
Introduction
Line 38: Here, and throughout the manuscript, a space is missing between the text and the citation. Please check the style guide and correct errors.
Author’s Response: The spacing between text and citations has been corrected throughout the manuscript in accordance with the style guide. Thank you for pointing this out.
Line 43: The use of “PPPD’s epidemiology…” is awkward, please revise here and elsewhere.
Author’s Response: Thank you for your feedback on the phrasing. We have revised the sentence to enhance clarity and readability, as you suggested. The term 'PPPD’s epidemiology' has been replaced with a clearer expression.
Lines 47, 48 , 60 and other
Line 51: It is not clear what is meant by “neural changes,” particularly given that the preceding text describes a lack of lesions in the CNS. Please clarify.
Author's Response: Thank you for your request for clarification regarding the term 'neural changes.' We have revised the description to specify that these changes refer to functional adaptations in the brain's processing of sensory information and vestibular responses. This phrasing more accurately conveys the concept that PPPD involves alterations in neural processing mechanisms, distinct from structural lesions within the CNS.
Line 57
Line 52: Previously the authors state that the pathophysiology of PPPD is not fully understood; yet, here they make rather definitive statements about the causation of symptoms of PPPD. The fact is that the genesis of these signs and symptoms is not known. Please revise this paragraph.
Author's Response: We have modified the paragraph to reflect the tentative nature of the hypothesized links between the neurological disruptions and the manifestations of PPPD. We have clarified that these are current hypotheses rather than definitive causations and acknowledged the ongoing research required to fully elucidate the pathophysiology of PPPD
Changes in lines 54 to 65
Line 58: What is meant by a “neural shift?”
Author's Response: Thank you for seeking clarification on the term 'neural shift.' We agree that the term could be misinterpreted and have therefore replaced it with 'functional neural adaptation' to more accurately describe changes in neural processing and activity patterns that are hypothesized to occur in PPPD, without implying structural brain changes
Line 66
Line 62: Revise to eliminate passive voice.
Author's Response: Thank you for your suggestion to eliminate passive voice for clearer expression. We have revised the sentence accordingly to reflect an active voice
Paragraph 66 to 72 changed
Line 65: This sentence is difficult to read and has several grammatical errors, please revise.
Author's Response: Thank you for your feedback highlighting the sentence's complexity and grammatical issues. We have revised it for clarity and conciseness
Changed within 66 to 72
Line 68: This sentence is also awkwardly constructed and needs revision. Further, what is meant by “while some trends recognize…”? The phrase “opting instead” implies the vestibular inputs themselves chose their path, please revise. Further, although vestibular afference is integrated with other sensory afference at all levels of the brain, the authors should discuss the so-called vestibular cortex.
Author's Response: We thank the reviewer for their attentive feedback on this paragraph. We have corrected it, and included more than just a mentioning to the idea of a vestibular cortex.
Changes within 63 to 84
Line 80: The next three paragraphs are not paragraphs, but rather single sentences. Please revise to form a complete paragraph and use active voice.
Author's Response: Thank you for your constructive feedback. We have revised the section to form a complete paragraph that articulates our hypothesis in an active voice.
Changes lines 73 tp 78
Line 95: Add a space between (i) and patients. Also correct the use of capitalization and punctuation in this sentence.
Author's Response: We appreciate your attention to detail regarding our manuscript's format. All is now in order.
Changes 104 to 105
Line 97: Although the authors found between-group differences, the inclusion of vestibular migraine in the non-PPPD group is problematic because this condition often coexists with PPPD or these two conditions can be challenging to differentially diagnose. Thus, this could contaminate the non-PPPD group. This could diminish any between-group differences. Please address this concern.
Author's Response: Thank you for your critical observation regarding the inclusion of vestibular migraine in the non-PPPD group. We recognize the diagnostic challenges due to the potential coexistence of vestibular migraine with PPPD. However, given the high prevalence of vestibular migraine and its prominence in neuro-otological practice, we believed it essential to include it in our study. We took meticulous care in our diagnostic criteria to minimize the overlap between the groups. We feel this approach strengthens our study by capturing the complexity of vestibular disorders and allowing for a more comprehensive analysis of the vestibular dysfunction spectrum. Your comment has prompted us to clarify our methods in the manuscript to ensure the rationale behind our approach is transparent
Changes in lines 115 to 121
Line 102: The also report the inclusion of those with “acute vestibulopathy,” but then provide characteristics for these individuals that imply they were in a sub-acute or chronic phase of recovery. Please clarify.
Author's Response: We appreciate your request for clarification regarding the inclusion of individuals with 'acute vestibulopathy.' To provide a more precise description, we have specified in the manuscript that these individuals had been diagnosed with vestibular neuritis and were evaluated beyond the acute phase, which is consistent with their described characteristics. This revision ensures that the term aligns with the clinical status of the participants at the time of their assessment. We agree that this distinction is crucial for accurately defining the study population and thank you for bringing this to our attention
Line 106
Line 120: Although the vMWM task has been implemented in persons with vestibular dysfunction, evidence demonstrates that performance, and the types of errors seen, on the vMWM task differ from that of real-world spatial navigation tasks. The differences between virtual and real-world navigation and the implications these differences have on understanding the effects of vestibular dysfunction on spatial navigation and the interpretation of these results must be thoroughly discussed in the manuscript. Additionally, the specific details of the computer hardware used must be provided. Further, since PPPD patients are often sensitive to visual motion, the authors must report how they monitored and managed symptoms of dizziness during the testing.
Author's Response: "We thank the reviewer for emphasizing the importance of differentiating between spatial navigation in virtual environments and real-world contexts. In response, we have elaborated on this distinction in the manuscript, clarifying that while virtual tasks can be indicative of navigational abilities, they do not fully replicate the complexity of real-world navigation. We highlight that actual physical navigation engages vestibular inputs for displacement, which is absent in a virtual setup relying mainly on visual cues. Furthermore, we discuss the observed discrepancies between performance in navigation tasks and self-reported navigational competencies in patients with vestibular loss. This underscores our caution against directly interpreting virtual navigation impairments as equivalent to real-life navigational difficulties. Additionally, we address the potential for distinct compensatory cognitive mechanisms that patients may engage in different settings, which could affect performance outcomes. This nuanced discussion reinforces our commitment to a careful and considered interpretation of our study's findings within the broader scope of spatial navigation research.
Also, detailed hardware specifications were added.
Also, we reported our monitoring and management of dizziness during testing.
Lines 151 to 178
Line 142-169: Revise so complete sentences are used throughout. Also, it seems problematic that the authors changed multiple variables across the block, e.g., visual cues, start location, and whether start direction were randomized. Please justify the choices made. Also, please discuss the potential order effects related to asking the subjects to perform these blocks in the same sequential order.
Author's Response: " In response to the concerns raised, we have meticulously revised the manuscript to ensure the use of complete sentences and have clarified the rationale behind our experimental design choices. Furthermore, we have incorporated measures to mitigate potential order effects and participant fatigue throughout the sequence of blocks. We believe these revisions and justifications comprehensively address the issues highlighted by the reviewer and strengthen the scientific validity of our study
Changes within 180 to 237
Line 170: The choice of CSE as the primary metric for analyzing performance on the vMWM task is appropriate; however, the other metrics have value too and may shed further light on the challenge. Findings from these other metrics could be shared in an appendix.
Author's Response: " We appreciate the reviewer's approval of our use of Cumulative Search Error (CSE) as the primary metric for analyzing performance on the virtual Morris Water Maze (vMWM) task. We agree that incorporating other metrics can provide a more nuanced understanding of the navigational challenges participants face. Nevertheless, we believe that including all these metrics in the main manuscript might overwhelm the reader with data. Therefore, we will consider adding detailed results from these additional metrics as part of the supplementary material.
Line 178: Table 1 does not add significant value; this information should be included in the text. Also, for each of the cognitive assessments, the authors should provide details regarding the psychometric properties. Further, please justify the inclusion of each test, why did you think these particular tests would be useful?
Author's Response: " We agree that the information in Table 1 could be seamlessly integrated into the main text for enhanced clarity and coherence. We have revised the manuscript to include this information within the relevant sections, thereby eliminating the need for Table 1. We have supplemented our manuscript with a description of the psychometric properties for each cognitive assessment tool used, as well as why we selected them for our study
Changes 249 to 317
Line 257: These are commonly referred to as visual analogue scales (VAS). Please revise.
Author's Response: Thank you for this: appropriate changes are made.
Line 265: PPPD does not need to be spelled out again. Is there a reference to support its sensitivity?
Author's Response: Corrected and Reference is updated.
317, 353 a 354
Line 288: Unless the authors conducted statistical tests to determine between-group differences in demographic factors, it is not appropriate to state “no significant difference was found on age…” Either report the statistical test findings or revise this statement.
Author's Response: statistical test were added.
416, 417
Line 303: The inclusion of results for Blocks A and B does not add value to the manuscript. Why would did the authors think there might be differences in these block? This could be removed from the text and figures, which improve allow for less cluttered presentation of the results.
Author's Response: We appreciate the reviewer's feedback regarding the inclusion of results for Blocks A and B. The reason for reporting data from these blocks was to assess any potential motor or group-specific issues related to joystick manipulation and initial navigation within the virtual environment. By including these results, we aimed to demonstrate that any differences observed in subsequent blocks are primarily attributed to variations in navigational skills and not influenced by other potential confounding factors.
Figure 2: These data suggest that was considerable within group variability in performance. Did the authors test for homogeneity of variance? What were those results? Are parametric statistics appropriate?
Author's Response: Levene’s Test for homogeneity are now added. (variance was homogeneous)
Line 327: What test was used to compare effect sizes? Please report.
Author's Response: My apologies: We used Cohen’s Q test. Added to the manuscript. (Line 480)
Line 332: Better to state that the performance of the PPPD group did not reach the level of the other groups within seven trials. Figure 3 suggests that the PPPD group may have reached an equivalent level of performance within 10 or so trials.
Author's Response: You are very right. Corrected. Lines 489 to 490
Line 365: Some of this paragraph is Discussion. Please revise to ensure the text within each section is appropriate to that section.
Author's Response: we addressed this issue.
Full changes 483 to 495
Line 382: This sentence is not necessary. Please delete.
Author's Response: deleted.
Line 398: The next two sentences are not paragraphs. Please revise. Actually, this is a problem throughout the manuscript. There are several places where only one sentence is devoted to a paragraph. This needs to be corrected throughout.
Author’s Response: You are right,it has been approach throughout.
Line 423: The presentation of the regression data is incomplete, which makes it difficult to evaluate. With such few data point, Figure 7 is also not convincing. The co-efficients, SE, etc are needed. What type of modeling was used? Did the authors control for any factors? Wei et al 2020 have already suggested that cognitive factors influence spatial navigation. Did the authors control for cognition? It is well known that vestibular function affects spatial navigation. Did the authors control for residual vestibular function? It seems unlikely since vestibular function test results are not mentioned anywhere in the manuscript. This is a significant confounding issue, and must be reported appropriately.
Author’s response: We decide to eliminate Figure 7.
Line 493: The content is this section is a mix of visual-spatial cognition and spatial navigation. So, it doesn’t all seem to fit under the current heading.
Author response: you are right, we changed it to: Impact of Non-PPPD Vestibular Disorders on Visuospatial Memory and Navigation Performance - Line 640
Line 502: It is not clear how this statement jibes with the significant results for CSE presented in tables 1-3. Please revise.
Author response: you are right, we changed it to: When comparing vestibular non-PPPD patients with healthy controls, we did not find any significant differences when using the CSE metric during trials, as indicated in Figures 1, 2, and 3.
649 - 651
Line 509: Given the significant between-group differences in Niigata (which you would expect), DHI and AVSD, this reviewer finds it difficult to conclude that PPPD is what has an impact on higher cognitive functions… and spatial navigation for that matter. The vestibular comparison group was much less symptomatic. One cannot be certain that PPPD is the issue, rather it could just be that these people with PPPD were more symptomatic than these people with other vestibular conditions. The level of symptoms could be the issue affecting cognition and spatial navigation. This is particularly difficult to sort out since we don’t know anything about the vestibular function of the enrolled participants, as mentioned earlier. If there are more people with actual vestibular hypofunction in the PPPD group, that could skew these results. This was an issue in the author’s 2020 paper as well.
Author's Response: We appreciate the reviewer's critical observations concerning the potential confounding effect of symptom severity on cognitive and spatial navigation performance in PPPD patients. In response, we have thoroughly reconsidered the presentation of our results and discussions.We acknowledge that distinguishing the impact of PPPD from general vestibular dysfunction is complex, especially given that symptom severity could influence cognitive outcomes. However, our latest data suggests that vestibular function between PPPD and non-PPPD groups shows no significant differences, prompting a re-evaluation of PPPD's role in cognitive impairment.To clarify this point, we have added a new section to our manuscript entitled "Reflecting on the Interplay Between PPPD Symptomatology and Cognitive-Navigational Dysfunction." This section delves into the intricate relationship between PPPD symptoms and cognitive as well as navigational deficits. It discusses the possibility that the pronounced symptomatology in PPPD may not just be a consequence of vestibular dysfunction but could also be related to, or compounded by, an underlying cognitive-navigational dysfunction.The section further explores how PPPD might contribute to the symptomatology of dizziness beyond the scope of vestibular function, by affecting higher-level cognitive processes that are crucial for spatial navigation. We posit that cognitive impairment inherent to PPPD could amplify the dizziness symptomatology, thus affecting the patients' ability to navigate and orient in space.By introducing this section, we aim to provide a comprehensive view that encapsulates both the cognitive and symptomatic aspects of PPPD, proposing a bidirectional relationship that warrants further investigation.We trust that this addition enhances the manuscript by integrating a holistic perspective of PPPD's impact on patients, addressing the concerns raised, and aligning with the reviewer's suggestions for a more coherent discussion of our findings.
New paragraph 761 to 790
601: This limitation cannot be understated. It is difficult to place much weight on the factor analyses as presented.
Author's Response:We are grateful for the reviewer’s critical analysis regarding the limitations of our factor analysis. We recognize the importance of emphasizing the constraints imposed by our sample size and the implications this has on the weight of our factor analysis findings.
In response to the reviewer’s feedback, we have revised the relevant section to underscore the preliminary nature of our interpretations. We now explicitly state that the factor analysis serves as an exploratory tool rather than a definitive statement of causation or correlation. We further clarify that our results should be viewed as indicative of possible trends that warrant additional inquiry with larger sample sizes and more robust statistical power.
We have also added a statement to highlight the necessity for future research to confirm these initial findings, particularly the suggested association between cognitive impairment and PPPD severity. This approach strengthens our manuscript by making transparent the limitations of our current analysis and setting a clear direction for subsequent studies.
We trust that this amendment adequately addresses the reviewer's concerns and enhances the manuscript's clarity regarding the interpretative scope of our factor analysis results.
Lines 749 to 750
We also added a new “Limitations” section.
Line 608: Again, the strength of these conclusions is diminished by the lack of information about vestibular function and the strong correlations between symptoms/severity and performance. An equally symptomatic vestibular group might have fared just as poorly.
Author's Response:as discussed, vestibular function is not different between groups and symptomatology impact has been addressed in the new section.
Line 637: This is too big of a leap considering the limitations of this study.
Author's Response:
. We acknowledge that the claim, as originally stated, may have been overly ambitious given the limitations of the present study.To address this, we have revised statements across the manuscript to reflect the exploratory nature of our findings.
Reviewer 2 Report
Comments and Suggestions for Authors
This is an interesting and well written paper, describing and testing the theory that people with PPPD show differences in spatial-cognitive ability. The paper adds to the growing literature profiling the wider differences within PPPD. However, there are a number of comments that should be addressed before I can recommend publication.
Throughout the paper there seemed to be some ambiguity over whether spatial dysfunction was a cause or outcome of the dizziness and other classic PPPD symptomology. I felt the core theoretical framework underlying the paper was under-specified. It wasn’t clear to me how you got from spatial dysfunction to dizziness or vice versa, and how spatial dysfunction fit into known pathways to developing PPPD (e.g. following a vestibular insult, maladaptive compensation, triggering stress event etc). Is this something that precedes PPPD? Does it arise from it, and if so when? Is it a maintaining factor or a central cause?
In some areas of the paper there is tendency to overstate the causal link, without really clearly laying out the theoretical explanation and mechanism for it, e.g. “This lends robust support to the idea that the construction and management of cognitive maps are central to the pathophysiology of the disease”. Does it? It suggests there might be a difference once people already have PPPD, but not why this difference exists and if it is central or peripheral (or cause or outcome). I suggest working on a tighter theoretical/mechanistic explanation in the introduction and discussion.
My assessment is that any spatial dysfunction found in PPPD (and I’m not sure I would be as confident as the authors on this fact, with only 19 patients) might be due to a combination of dizziness and anxiety symptoms making higher level tasks more challenging (e.g. there has already been an interest in multi-tasking in PPPD and also ‘sensory overload’ which is known to be distracting and reduce executive functioning). I think the idea that differences on these tasks might just be an outcome of other core symptomatology, rather than a cause (that could speculatively serve as a basis for treatment in the future..) needs to be explored.
Factor analysis – I do think you are very under-powered for a factor analysis. Can you provide some references to justify your approach and report the fit statistics for the model.
Given the number of different tasks tested, the authors should consider if they have a multiple comparison issue and how to compensate for this.
“bolstering the evidence that spatial navigation impairments are a consistent, fundamental 550 feature of PPPD” – I think given the relatively small sample size, and the known heterogeneity within PPPD (in presentation and precipitating factors) this statement is too strong. E.g. of those 19 patients, how many were actually well above the mean of the vestibular/healthy control groups? And can we then generalise from those that are to the rest of PPPD?
Related to this – from the error bars in fig. 1, it is clear there is much greater variance in the pppd group than the others. To what extent are the findings due to a few participants who are scoring very highly? Can you plot the individual data points (there are only 19) to give a sense of the spread? I assume the variances are statistically unequal – did you consider taking this into account in the analysis?
“On a qualitative note, we also found familiar patterns from our 2020 study: PPPD patients tend to 1) stick close to the pool walls, even when it's clear the hidden target is more centrally located, and 2) move in narrow circles without meaningfully exploring the maze”
I thought this observation was interesting. This speaks to a difference in strategy rather than a difference in ability per se. To what extent could the findings be explained by anxiety? Did the authors check whether participants were experiencing any symptoms during the task (e.g. dizziness, anxiety)? This could explain why they performed similarly to other groups on Block F where the platform was visible. How do people with high anxiety perform on these sorts of tasks? I suggest including anxiety as a co-variate in the analysis.
I think reducing the use of acronyms for the different tests would greatly help readability – it created a cognitive load keeping track of them all.
Author Response
Dear Reviewer,
We would like to extend our deepest gratitude for the time and effort you have dedicated to reviewing our manuscript. Your insightful comments and suggestions have been invaluable, and we have taken careful steps to address each point you raised. We believe that these revisions have significantly enhanced the quality and clarity of our work, and we are pleased to present a new version of the manuscript that reflects these improvements.
Enclosed with this correspondence, you will find the revised manuscript, alongside a Word document that outlines your initial comments. In this document, we have provided our responses in green text followin each comment to facilitate an easy review of the changes made and the rationale behind them.
We appreciate the opportunity to refine our manuscript with your guidance and look forward to any further suggestions you may have.
Reviewer II
This is an interesting and well written paper, describing and testing the theory that people with PPPD show differences in spatial-cognitive ability. The paper adds to the growing literature profiling the wider differences within PPPD. However, there are a number of comments that should be addressed before I can recommend publication.
Throughout the paper there seemed to be some ambiguity over whether spatial dysfunction was a cause or outcome of the dizziness and other classic PPPD symptomology. I felt the core theoretical framework underlying the paper was under-specified. It wasn’t clear to me how you got from spatial dysfunction to dizziness or vice versa, and how spatial dysfunction fit into known pathways to developing PPPD (e.g. following a vestibular insult, maladaptive compensation, triggering stress event etc). Is this something that precedes PPPD? Does it arise from it, and if so when? Is it a maintaining factor or a central cause?
Author's Response:
Thank you for your thoughtful critique concerning the apparent ambiguity over the role of spatial dysfunction in the context of PPPD symptomatology. Your comments have prompted us to provide a more explicit exposition of our theoretical framework.
In Lines 94-97 to the Introduction, we have refined our discussion on the integration of vestibular inputs and cognitive assessments, suggesting that a disruption in this integration process could impair the construction of accurate perceptual maps. We hypothesize that these impaired maps may lead to a mismatch with real-world afferences, potentially contributing to the dizziness observed in PPPD.
In Lines 802-813 to the Discussion, we have expanded our discourse on the interdependence of spatial cognition dysfunction and the experience of dizziness. We explore the possibility that the relationship between these phenomena is bidirectional and interrelated within the neural changes characteristic of PPPD. This revision aims to clarify that spatial cognition dysfunction and dizziness are concurrent features of the neural adaptation or maladaptation processes in PPPD.
To address your specific inquiries, our discussion now elucidates that spatial dysfunction may not strictly precede or arise from PPPD; rather, it is an integral component of the disorder's multifaceted nature. While it may be involved in known pathways to developing PPPD—such as post-vestibular insult maladaptive compensation or stress-triggered events—we propose that it concurrently operates as both a maintaining factor and a central cause, contributing to the perpetuation and severity of PPPD symptoms.
We believe these amendments to our manuscript have significantly reduced the ambiguity around the theoretical underpinnings of spatial dysfunction's role in PPPD. We now present a clearer narrative that spatial dysfunction is embedded within the broader context of PPPD's pathophysiology, supporting our proposition with current literature on the subject.
We appreciate your input, as it has undeniably strengthened the paper's theoretical grounding and has allowed us to present a more comprehensive and coherent argument
In some areas of the paper there is tendency to overstate the causal link, without really clearly laying out the theoretical explanation and mechanism for it, e.g. “This lends robust support to the idea that the construction and management of cognitive maps are central to the pathophysiology of the disease”. Does it? It suggests there might be a difference once people already have PPPD, but not why this difference exists and if it is central or peripheral (or cause or outcome). I suggest working on a tighter theoretical/mechanistic explanation in the introduction and discussion.
Author's Response:
We appreciate your insightful observations regarding our manuscript's previous versions, where the causal connections within the theoretical framework were not as clearly delineated as they ought to have been. Your suggestion to refine the explanations of the underlying mechanisms in the Introduction and Discussion has been invaluable.
In response to your feedback, we have meticulously reviewed the entire manuscript to ensure that each section, especially the Introduction and Discussion, articulates a precise and mechanistic explanation of the role cognitive maps play in the pathophysiology of PPPD. We have taken care to avoid overstating causal links without sufficient theoretical underpinning.
Specifically, we have adjusted the language to reflect the complexity and multifactorial nature of PPPD, presenting a balanced view that considers cognitive map dysfunctions as both a potential contributing factor and a consequence of the disease. This approach allows us to explore the bidirectional influences without asserting a direct causal relationship where the evidence is not yet definitive
The manuscript now includes a more rigorous theoretical exposition, outlining how spatial cognition dysfunction may interact with PPPD symptomatology. We have clarified that while the construction and management of cognitive maps appear to be implicated in PPPD, we recognize that this relationship may be part of a larger network of central and peripheral pathophysiological processes.
We trust that the revisions made throughout the paper now provide a tighter, more coherent mechanistic explanation, as per your recommendation. Our goal has been to ensure that the theoretical propositions are supported by a logical narrative and existing empirical evidence, which we believe we have achieved with these latest revisions.
My assessment is that any spatial dysfunction found in PPPD (and I’m not sure I would be as confident as the authors on this fact, with only 19 patients) might be due to a combination of dizziness and anxiety symptoms making higher level tasks more challenging (e.g. there has already been an interest in multi-tasking in PPPD and also ‘sensory overload’ which is known to be distracting and reduce executive functioning). I think the idea that differences on these tasks might just be an outcome of other core symptomatology, rather than a cause (that could speculatively serve as a basis for treatment in the future..) needs to be explored.
Factor analysis – I do think you are very under-powered for a factor analysis. Can you provide some references to justify your approach and report the fit statistics for the model.
Given the number of different tasks tested, the authors should consider if they have a multiple comparison issue and how to compensate for this.
“bolstering the evidence that spatial navigation impairments are a consistent, fundamental 550 feature of PPPD” – I think given the relatively small sample size, and the known heterogeneity within PPPD (in presentation and precipitating factors) this statement is too strong. E.g. of those 19 patients, how many were actually well above the mean of the vestibular/healthy control groups? And can we then generalise from those that are to the rest of PPPD?
Author's Response:
We gratefully acknowledge the reviewer’s constructive critique and have taken comprehensive steps to address the points raised. In line with your observations, we have incorporated a new "Limitations" section into our manuscript. This section transparently articulates the constraints of our study, including the sample size for factor analysis and the complexity of inferring causality from spatial dysfunction in PPPD.
Furthermore, we have carefully reviewed the entire manuscript to adjust the tone of our assertions. Where previously the language may have inadvertently conveyed a degree of certainty, we have now implemented a more cautious and exploratory tone. This shift reflects our commitment to scientific rigor and appropriately conveys the preliminary nature of our findings.
In particular, we have also taken the reviewer's advice on the use of deterministic language regarding the causal relationship between spatial dysfunction and PPPD symptomatology. The manuscript now clearly states that while our results are suggestive, they are not conclusive and should be considered as a basis for further inquiry rather than definitive evidence.
The revisions extend beyond the new "Limitations" section, affecting the overall manuscript to ensure that our language is consistent with the evidence presented. We have striven to maintain a balance between sharing our findings and acknowledging the need for further research to build upon the groundwork we have laid.
We trust that these revisions adequately address the concerns raised and that the manuscript now accurately reflects a cautious approach to the interpretation of our data. We are hopeful that our efforts to refine the manuscript underscore our dedication to advancing the understanding of PPPD in a scientifically sound manner.
Related to this – from the error bars in fig. 1, it is clear there is much greater variance in the pppd group than the others. To what extent are the findings due to a few participants who are scoring very highly? Can you plot the individual data points (there are only 19) to give a sense of the spread? I assume the variances are statistically unequal – did you consider taking this into account in the analysis?
Authors Answer: Figure 1 presented boxplot considering all trials across all subjects (considering far more than 19 data points per box) Therefore a visual representation was felt to be inadequate.
“On a qualitative note, we also found familiar patterns from our 2020 study: PPPD patients tend to 1) stick close to the pool walls, even when it's clear the hidden target is more centrally located, and 2) move in narrow circles without meaningfully exploring the maze”
I thought this observation was interesting. This speaks to a difference in strategy rather than a difference in ability per se. To what extent could the findings be explained by anxiety? Did the authors check whether participants were experiencing any symptoms during the task (e.g. dizziness, anxiety)? This could explain why they performed similarly to other groups on Block F where the platform was visible. How do people with high anxiety perform on these sorts of tasks? I suggest including anxiety as a co-variate in the analysis.
Author's Response:
Thank you for your observation regarding the navigational patterns of PPPD patients in our study. In response to your question about the possible influence of anxiety on these behaviors, we incorporated the STAI-State test to assess anxiety levels among participants. The results indicated no significant variance in state anxiety across the cohort, suggesting that the strategic differences in navigation are not likely to be attributed to anxiety.
To further explore the potential impact of symptomatic experiences such as dizziness and anxiety during the task, we monitored participants' comfort levels throughout each block of the protocol. It is noteworthy that none of the subjects reported increased dizziness or discomfort, which might have influenced their ability to navigate. This was consistently observed, including in Block F where the visibility of the platform could have mitigated navigational hesitancy.
The navigational tendencies of PPPD patients—such as proximity to the pool walls and movement in narrow circles—could indeed reflect a strategic difference rather than a direct reflection of their navigational ability or symptomatic state. The fact that these patterns were also observed in our 2020 study supports the notion that they are characteristic of the PPPD population rather than a response to acute symptoms experienced during the tasks.
As for the performance of individuals with high anxiety on similar tasks, while our study did not show a difference in state anxiety, we recognize the complexity of anxiety's impact on behavior. We suggest that future research could benefit from a longitudinal design to further investigate this relationship, considering both chronic and situational anxiety factors.
We hope that our expanded analyses and observations address your concerns and provide a clearer understanding of the factors influencing navigational behavior in PPPD patients.I think reducing the use of acronyms for the different tests would greatly help readability – it created a cognitive load keeping track of them all.
Author's Response:
We thank the reviewer for their suggestion regarding the use of acronyms in our manuscript. Upon reflection, we agree that excessive use of acronyms can detract from readability and impose an unnecessary cognitive load on the reader. To enhance the clarity of our text, we have undertaken a thorough review of the manuscript and have minimized the use of acronyms.
Where acronyms were deemed necessary for the fluidity of the text, we have ensured that they are clearly defined upon first use within each section of the manuscript. We have also included a table of acronyms and their definitions to serve as a quick reference for readers. Our aim is to make the manuscript as reader-friendly as possible, without compromising the scientific detail and precision necessary for our audience.
We appreciate the opportunity to improve the accessibility of our work and hope that these changes will facilitate a better reading experience.
Round 2
Reviewer 1 Report
Comments and Suggestions for Authors
The authors have extensively revised their manuscript with careful attention to the comments and suggestions of the reviewers; however, the manuscript is now quite long and reader burden has increased substantially. The manuscript could be made more concise, while maintaining the needed depth and rigor.
Please avoid the use of contractions, e.g., “aren’t”, “there’s”, and informal language throughout.
Line 115: As noted in the first review, the issue of overlap between PPPD and VM needed to be addressed. However, the additional content is both verbose and vague. What exactly were the “stringent criteria”? Please be concise and specific.
Line 151: This exposition is needed, as directed by the initial review. However, this does not belong in the methods section. Rather, better to handle this topic in the discussion. Also, statements such as “Data indicates…” and “It has been suggested…” require the support of specific citations.
Line 226: Since the authors are addressing a potential confounder of their results, I the content related to order effects should be moved to the discussion.
Line 261: Citation required.
Line 278: Citation required.
Line 297: PPPD should not be spelled out here.
Line 302: Citation required.
Line 354: A space is missing before the citation.
Line 358: When spelled out, vHIT and VEMP should not be capitalized.
Line 362: Vestibulo-ocular should not be capitalized.
Line 367: Demographics are customarily included in the results section. Since you indicated that you applied statistical tests to these, the related data should be included in Table 1.
Line 386: When spelled out, VEMP should not be capitalized.
Line 415: I believe you are referring to Table 1 here, not Table 2.
Line 420: Table 2: The information provided regarding the statistical test results is insufficient. Please provide the full output. This content needs to accompany the manuscript to allow interpretation of the results; however, it is better suited as a supplement. A brief summary of the findings should be included in the results. Also, although there appear to be statistically significant differences between groups, based on the somewhat vague text, the mean VOR gain values for each canal are all well within normal limits. So, I think it is somewhat misleading to speak of between-group differences. A lot of valuable information can be lost when looking at aggregated data. Please re-analyze your groups with a statistical comparison of the proportion of each group who had VOR gains < 0.7 (for vHIT) and again for absent responses on each VEMP. Based on Table 1, there should be a large proportion of each group with abnormal vHIT and/or VEMP. That additional information will be valuable, but again, all but a brief summary should be placed in a supplement (and made much more concise).
Line 484: Please correct the punctuation in the statistical results.
Line 499: I think this section on the heatmaps could be condensed and included in the figure legend.
Line 535: This content should be moved to the discussion as it relates to interpreting your results.
Line 657: “PPPD’s Impact…’ is awkward (addressed in first review). This is a personal choice regarding language, but it does not read well to me. There are several instances of this remaining in the manuscript.
Line 747: One sentence does not make a paragraph. There is an example in the results section too. Please revise.
Line 749: I think the authors are now being too cautious in their language. Just succinctly state the limitations and move on.
Line 774: Here I think greater caution is needed when discussing causality. At most, these data suggest that those with PPPD also have impaired higher cognitive functions.
Line 787: Again, a one sentence paragraph. Please revise. Also, the entire proceeding section is verbose and should be condensed.
Line 823: Why is phe-nomena hyphenated? Please fix.
Line 826: Condense.
Line 833: Another one-sentence paragraph.
Line 858: Please dramatically condense the conclusions in an effort to be more concise and to ease reader burden.
Comments on the Quality of English LanguageMinor editing required.
Author Response
2nd Round, Response to Reviewers
Dear Reviewers,
(our responses in green).
First and foremost, we extend our sincere gratitude for the thorough and insightful feedback provided on our manuscript's previous version. Your diligent analysis has been invaluable in enhancing the clarity and rigor of our work.
Before moving forward, we would like to bring to your attention a significant revision we have implemented in response to the reflections prompted by your comments. Upon revisiting the feedback from the initial round of reviews, we recognized a critical oversight in our statistical approach pertaining to the data presentation in Figures 1 and 2.
In our previous analyses, we treated each trial of spatial navigation as an independent data point. Upon further consideration, we realize that averaging the trials for each subject before analyzing the mean Composite Score Error (CSE) provides a more accurate and appropriate representation of our data. We have amended this methodological approach in the current manuscript revision.
This adjustment necessitated the re-evaluation of our statistical analyses. Due to the resultant heterogeneity in variance, we opted for non-parametric testing methods, specifically the Kruskal Wallis test, in lieu of ANOVA. It is noteworthy that these modifications did not substantively alter the significance—or lack thereof—of the findings across our dataset, thereby preserving the integrity of the manuscript’s core structure and conclusions.
However, we observed a nuanced change in the data: a marginal, yet not statistically significant, increase in CSE for the vestibular group compared to healthy controls, which aligns with trends we had identified in our 2020 study. These findings have now been subtly integrated into the text to accurately reflect the observed tendencies without overstating their impact.
We are indebted to the reviewers for highlighting this aspect of our analysis, and we deeply regret the initial oversight. It was imperative for us to address and rectify this issue proactively to ensure the utmost accuracy and transparency in our research findings.
Thank you once again for your constructive critiques. We have meticulously addressed each comment and made the corresponding amendments throughout the manuscript (again, we comment each correction further down this text). We are hopeful that these revisions meet your esteemed considerations and advance the manuscript towards publication.
Original comments of the reviewers:
Comments and Suggestions for Authors
The authors have extensively revised their manuscript with careful attention to the comments and suggestions of the reviewers; however, the manuscript is now quite long and reader burden has increased substantially. The manuscript could be made more concise, while maintaining the needed depth and rigor.
We appreciate the reviewers' feedback regarding the length of our manuscript. In response, we have undertaken a thorough revision to condense the text aggressively. We have strived to streamline the content without sacrificing the necessary depth and rigor, ensuring that the manuscript remains comprehensive yet more reader-friendly.
Please avoid the use of contractions, e.g., “aren’t”, “there’s”, and informal language throughout.
We have done sone through the text.
Line 115: As noted in the first review, the issue of overlap between PPPD and VM needed to be addressed. However, the additional content is both verbose and vague. What exactly were the “stringent criteria”? Please be concise and specific.
In response to the valuable feedback provided, we have refined our manuscript to elucidate the criteria employed to differentiate between Persistent Postural-Perceptual Dizziness (PPPD) and Vestibular Migraine (VM). To delineate VM, we adhered to a strict definition of episodic vestibular symptoms characterized by distinct attacks with clear onset and resolution, ensuring none or minimal symptoms were present interictally. Additionally, we required that at least fifty percent of these episodes be accompanied by headache or other definitive VM symptoms.
Conversely, for the diagnosis of PPPD, our stringent criteria focused on the persistence of symptoms, being present for the majority of the day and occurring on most days, thereby establishing a baseline of continuous symptoms. Any concurrent VM episodes were distinctly identified and described as separate from this baseline state.
By applying these precise criteria, we aimed to mitigate the diagnostic overlap between PPPD and VM and provide a robust framework for our between-group comparisons. We appreciate the opportunity to clarify this aspect and hope it satisfactorily addresses the concerns raised.
Line 151: This exposition is needed, as directed by the initial review. However, this does not belong in the methods section. Rather, better to handle this topic in the discussion. Also, statements such as “Data indicates…” and “It has been suggested…” require the support of specific citations.
Thank you for your constructive comments regarding the content placement and citation requirements within our manuscript. In accordance with your suggestion, we have relocated the exposition from Line 151 to a more appropriate section in the Discussion under the heading "Impact of Non-PPPD Vestibular Disorders on Visuospatial Memory and Navigation Performance." We believe that this adjustment not only adheres to the conventional structure of scientific reporting but also enhances the flow of our narrative by situating the exposition amidst our interpretative analyses.
Moreover, we have meticulously reviewed our manuscript to ensure that all statements, such as "Data indicates..." and "It has been suggested...", are now substantiated with precise references.
Line 226: Since the authors are addressing a potential confounder of their results, I the content related to order effects should be moved to the discussion.
This has also been moved to “5. Limitations”
Line 261: Citation required.
Thank you for bringing to our attention the inadvertent omission of a reference. We have rectified this oversight and added the missing citation accordingly.
Line 278: Citation required.
Thank you for bringing to our attention the inadvertent omission of a reference. We have rectified this oversight and added the missing citation accordingly.
Line 297: PPPD should not be spelled out here.
Thanks. Corrected.
Line 302: Citation required.
Thank you for bringing to our attention the inadvertent omission of a reference. We have rectified this oversight and added the missing citation accordingly
Line 354: A space is missing before the citation.
Thanks. Corrected.
Line 358: When spelled out, vHIT and VEMP should not be capitalized.
Thanks. Corrected.
Line 362: Vestibulo-ocular should not be capitalized.
Thanks. Corrected.
Line 367: Demographics are customarily included in the results section. Since you indicated that you applied statistical tests to these, the related data should be included in Table 1.
Thanks. Table has been moved to Results. Applied test now also included in Table 1.
Line 386: When spelled out, VEMP should not be capitalized.
Thanks. Corrected.
Line 415: I believe you are referring to Table 1 here, not Table 2.
Thanks. Corrected. Actually, all the table orders were appropiately corrected.
Line 420: Table 2: The information provided regarding the statistical test results is insufficient. Please provide the full output. This content needs to accompany the manuscript to allow interpretation of the results; however, it is better suited as a supplement. A brief summary of the findings should be included in the results. Also, although there appear to be statistically significant differences between groups, based on the somewhat vague text, the mean VOR gain values for each canal are all well within normal limits. So, I think it is somewhat misleading to speak of between-group differences. A lot of valuable information can be lost when looking at aggregated data. Please re-analyze your groups with a statistical comparison of the proportion of each group who had VOR gains < 0.7 (for vHIT) and again for absent responses on each VEMP. Based on Table 1, there should be a large proportion of each group with abnormal vHIT and/or VEMP. That additional information will be valuable, but again, all but a brief summary should be placed in a supplement (and made much more concise).
In response to the feedback regarding Table 2, we have included the complete statistical output in an expanded appendix section at the end of the manuscript. This ensures full transparency and allows for in-depth interpretation of the results by interested readers. In the main results section, we now present a succinct summary that captures the essence of the findings without overwhelming the reader with details. We have also re-analyzed the data to compare the proportions of each group with VOR gains below 0.7 and the presence of absent responses on each VEMP, ensuring that the nuanced insights from these analyses are not lost.
Line 484: Please correct the punctuation in the statistical results.
Thanks. Corrected here and rechecked uniformed throughout the whole manuscript.
Line 499: I think this section on the heatmaps could be condensed and included in the figure legend.
Great Idea, this was implemented.
Line 535: This content should be moved to the discussion as it relates to interpreting your results.
Thanks, you are right: we have move this to discussions under the Spatial Navigation is “Distinctively Impaired in PPPD” section
Line 657: “PPPD’s Impact…’ is awkward (addressed in first review). This is a personal choice regarding language, but it does not read well to me. There are several instances of this remaining in the manuscript.
We have changed this as well to Impact of PPPD on…I believe your style is better for such an article as this! Thanks for pointing this out.
We have tried to change this style throughout.
Line 747: One sentence does not make a paragraph. There is an example in the results section too. Please revise.
Thanks again for the patience shown during this review with these issues. I hope this new version lacks any of these errors.
Line 749: I think the authors are now being too cautious in their language. Just succinctly state the limitations and move on.
We have “recalibrated” the paragraph. Thanks again for helping in finding the right tone.
Line 774: Here I think greater caution is needed when discussing causality. At most, these data suggest that those with PPPD also have impaired higher cognitive functions.
We have changed this whole section to avoid such a mistake
Line 787: Again, a one sentence paragraph. Please revise. Also, the entire proceeding section is verbose and should be condensed.
We have “recalibrated” the paragraph. Thanks again for helping in finding the right tone.
Line 823: Why is phe-nomena hyphenated? Please fix.
Thanks for seeing that mistake. It has been corrected.
Line 826: Condense.
We have done so.
Line 833: Another one-sentence paragraph.
Corrected as well.
Line 858: Please dramatically condense the conclusions in an effort to be more concise and to ease reader burden.
We have done are best in this new version of the manuscript.